# Real-Time Epidemiology and Acute Care Need Monitoring and Forecasting for COVID-19 via Bayesian Sequential Monte Carlo-Leveraged Transmission Models

**DOI:** 10.3390/ijerph21020193

**Published:** 2024-02-07

**Authors:** Xiaoyan Li, Vyom Patel, Lujie Duan, Jalen Mikuliak, Jenny Basran, Nathaniel D. Osgood

**Affiliations:** 1Department of Computer Science, University of Saskatchewan, Saskatoon, SK S7N 5C9, Canada; vyom.patel@usask.ca (V.P.); lujie.duan@usask.ca (L.D.); jalen.mikuliak@usask.ca (J.M.); nathaniel.osgood@usask.ca (N.D.O.); 2Saskatchewan Health Authority, Saskatoon, SK S7K 0M7, Canada; jenny.basran@saskhealthauthority.ca

**Keywords:** COVID-19, particle filtering, machine learning, epidemiologic modeling, compartmental model, projection and intervention

## Abstract

COVID-19 transmission models have conferred great value in informing public health understanding, planning, and response. However, the pandemic also demonstrated the infeasibility of basing public health decision-making on transmission models with pre-set assumptions. No matter how favourably evidenced when built, a model with fixed assumptions is challenged by numerous factors that are difficult to predict. Ongoing planning associated with rolling back and re-instituting measures, initiating surge planning, and issuing public health advisories can benefit from approaches that allow state estimates for transmission models to be continuously updated in light of unfolding time series. A model being continuously regrounded by empirical data in this way can provide a consistent, integrated depiction of the evolving underlying epidemiology and acute care demand, offer the ability to project forward such a depiction in a fashion suitable for triggering the deployment of acute care surge capacity or public health measures, and support quantitative evaluation of tradeoffs associated with prospective interventions in light of the latest estimates of the underlying epidemiology. We describe here the design, implementation, and multi-year daily use for public health and clinical support decision-making of a particle-filtered COVID-19 compartmental model, which served Canadian federal and provincial governments via regular reporting starting in June 2020. The use of the Bayesian sequential Monte Carlo algorithm of particle filtering allows the model to be regrounded daily and adapt to new trends within daily incoming data—including test volumes and positivity rates, endogenous and travel-related cases, hospital census and admissions flows, daily counts of dose-specific vaccinations administered, measured concentration of SARS-CoV-2 in wastewater, and mortality. Important model outputs include estimates (via sampling) of the count of undiagnosed infectives, the count of individuals at different stages of the natural history of frankly and pauci-symptomatic infection, the current force of infection, effective reproductive number, and current and cumulative infection prevalence. Following a brief description of the model design, we describe how the machine learning algorithm of particle filtering is used to continually reground estimates of the dynamic model state, support a probabilistic model projection of epidemiology and health system capacity utilization and service demand, and probabilistically evaluate tradeoffs between potential intervention scenarios. We further note aspects of model use in practice as an effective reporting tool in a manner that is parameterized by jurisdiction, including the support of a scripting pipeline that permits a fully automated reporting pipeline other than security-restricted new data retrieval, including automated model deployment, data validity checks, and automatic post-scenario scripting and reporting. As demonstrated by this multi-year deployment of the Bayesian machine learning algorithm of particle filtering to provide industrial-strength reporting to inform public health decision-making across Canada, such methods offer strong support for evidence-based public health decision-making informed by ever-current articulated transmission models whose probabilistic state and parameter estimates are continually regrounded by diverse data streams.

## 1. Introduction

A novel coronavirus and accompanying infectious disease were reported to the World Health Organization (WHO) in Wuhan, China in December of 2019 [1]. The WHO declared this outbreak a Public Health Emergency of International Concern in January of 2020, designating this new coronavirus disease COVID-19 [1]. Global travel and endogenous spread across hundreds of countries have yielded a worldwide pandemic, with rapidly rising totals of over 752 million confirmed cases and over 6.8 million confirmed deaths through 30 January 2023 [2].

During the COVID-19 pandemic, ongoing public health order planning and replanning associated with rolling back and reinstituting measures and conducting timely messaging has benefited from the availability of empirical time series—often holding evidence of shifts in epidemiology, availability of acute care resources, and changes in behaviour with regards to risk, testing, vaccination uptake, and clinical presentation. At the same time, decision-making has relied heavily on a variety of types of dynamic models.

Several previous studies [3,4,5,6,7] showed success in monitoring, estimating, and predicting the transmission of infectious diseases by stochastic filtering of mathematical epidemiology models using observed datasets via sequential Monte Carlo (SMC) machine learning algorithms. SMC methods were introduced in the early 2000s [8,9], and commonly go by the name of particle filtering (PF). Such studies have demonstrated that projections forward from dynamic models in health and health care offer substantial additional value if they can be informed by up-to-date, grounded estimates of the current situation. The particle filtering method—together with several variants—has also been used for COVID-19 [10,11,12,13,14,15] in the last two years since this new infectious disease emerged. Most of these studies used public health surveillance data—such as daily reported cases and daily hospitalized admission patients—to track the transmission dynamics. After the SARS-CoV-2 virus was confirmed detected in untreated wastewater [16,17,18,19,20,21,22,23,24,25], several researchers [14] used wastewater surveillance data to ground the mathematical epidemiology models via a partially observed Markov processes (POMP) model. These methods use Markov chain Monte Carlo and sequential Monte Carlo (particle filtering) methods.

This paper presents the use of PF with a model deployed by the health system and used internally for routine provincial-level reporting and decision-making since the fourth month of the pandemic. Such PF incorporated a COVID-19 compartmental transmission model and a wide variety of observed daily datasets from both public health surveillance data and wastewater surveillance data. Within this context, the COVID-19 model provides an integrated characterization of disease transmission, a natural history of infection including both frankly symptomatic and oligo-/pauci-symptomatic pathways, distinct passive and active case-finding systems for the occurrence of travelling cases, basic COVID-19 related acute care flows and occupancy, characterization of two dose-specific vaccination stages, and mortality. Important model outputs include estimates (via sampling) of the effective reproductive number, the count of undiagnosed infectives, and the count of individuals at different stages of the natural history of infection along both pathways. Since July 2020, the model further incorporated a representation of SARS-CoV-2 fecal viral shedding, and when wastewater evidence is available, the PF framework makes use of a likelihood term comparing the empirical viral concentration of SARS-CoV-2 in wastewater with model expectations for that concentration.

The model was built in concert with the Saskatchewan Health Authority and has been in production use for regular health system reporting since June 2020, with some model findings informing understanding of the evolving epidemiological context as early as April 2020. Since that time, and beyond its use for reporting to the Saskatchewan Health Authority and Saskatchewan Ministry of Health, the model has been used to deliver reporting contracts with the Public Health Agency of Canada (for each Canadian province), First Nations and Inuit Health Branch (FNIHB). The resulting reports have proven particularly key in day-to-day instituted health system reporting and informing planning for the Canadian midwestern province of Saskatchewan. In this paper, we characterize the structure of the model and present the results of applying the model to the population of the city of Saskatoon in the province of Saskatchewan, Canada, during the period of wild type SARS-CoV-2 and the alpha variant [26] from 22 February 2020 to 31 July 2021.

## 2. Methods

### 2.1. Deterministic Compartmental Model

We describe here the compartmental model used within this system, which characterizes the total population as divided into different compartments distinguished by different pathways of natural progression, severity of illness, diagnosis, and acute care use. For simplicity, our description of the model omits discussion of the evolution of that model, pausing only to note that the vast majority of the model as described here was in use at the start of regular reporting in June 2020. We also exclude from this section a characterization of variants of that model, differing particularly in the levels of stratification involved. We further exclude discussion of variant-specific adjustment of values of some parameters otherwise treated as constant and the model structure adjusted to accommodate further variants in the application.

The structure of the COVID-19 compartmental model is shown in Figure 1 and employs a time unit of days. The compartments of the model are introduced as follows. The model contains a largely orthogonal characterization of progression along two possible natural histories of infection (on one hand) and diagnosis status (on the other). Specifically, the model dichotomizes both the infective (compartments denoted by names prefixed by *I* or *H*) and recovered (compartments prefixed by *R*) populations into diagnosed (subscripted by${\;}_{D}$) and undiagnosed (subscripted by${\;}_{U}$) status, depending on whether an individual has been diagnosed via lab-confirmed PCR testing. The infective population in the model was further divided into two groups: hospitalized individuals (compartments $H_{NICU}$ and $H_{ICU}$) and those in the community (subcompartments of the supercompartment *I*). Supercompartment *I* of infectives in the community is characterized by dividing it into three groups based on the stage of the natural history of infection—presymptomatics (compartments $I_{AU}$ and $I_{AD}$) and those at a later stage along each of the two parallel pathways of infection distinguished by degree of symptomaticity. Specifically, the model treats infected individuals as proceeding from the (infectious) presymptomatic phase to one of two possible natural histories of infection: a frankly symptomatic pathway and an oligosymptomatic route of progression, which accept fractions $1-f_{PA}$ and $f_{PA}$, respectively, of undiagnosed individuals proceeding from presymptomatic compartment $I_{AU}$. The frankly symptomatic pathway starts at an early stage in which individuals have not yet had the opportunity to exhibit complications (compartments $I_{YU}$ and $I_{YD}$) and symptoms are assumed to be mild. The progression of an individual from the first to the second symptomatic stage marks the point where any complications emerge, with a specified fraction (denoted as $f_{H}$) of progressing individuals (regardless of erstwhile diagnosis status) developing severe or critical complications. Such individuals suffering complications are presumed to lead to presentation for care and hospitalization. Frankly symptomatic individuals absent complications proceed on to a stage involving symptomatic individuals beyond the risk of complications (compartments $I_{YNU}$ and $I_{YND}$). In contrast to the frankly symptomatic pathway, the oligosymptomatic pathway proceeds from the presymptomatic stage through a natural history of infection in which infected individuals remain infective but never develop symptoms sufficient to motivate care-seeking; compartments along this pathway are denoted by an${\;}_{A}$ subscript. Like their symptomatic counterparts, oligosymptomatic infectives are characterized as proceeding through two subsequent compartments of $I_{A}$, with the timing of progression identical to the frankly symptomatic stages—oligosymptomatic stage 1 (compartments $I_{A2U}$ and $I_{A2D}$) and oligosymptomatic stage 2 (compartments of $I_{A3U}$ and $I_{A3D}$). The model also considers the vaccinated population, where only susceptible individuals are assumed to be administered vaccines. Compartment V1 represents the persons who have only received one dose of a COVID-19 vaccine, and V2 represents the persons who have received two vaccine doses. As is detailed further below, vaccinated individuals are treated as remaining subject to some vaccine-efficacy-moderated risk of infection (denoted $e_{1}$ for only having one dose and $e_{2}$ for having two doses).

#### 2.1.1. Diagnosis and Case Finding

In this COVID-19 compartmental model, infected patients can be diagnosed both by passive case finding via presentation for care and (separately) via active case finding, such as through contact tracing, screening, and mass testing [27]. Passive case finding is treated as diagnosing symptomatic infectives who present for care and is treated as endogenously driven within the model. Such presentation-driven diagnosis is represented by red flows in Figure 1 and proceeds from compartments of undiagnosed symptomatic infectives that have not yet exhibited complications $I_{YU}$ to the next stage compartment of diagnosed individuals $I_{YND}$. In contrast, reflecting the fact that active case finding can identify individuals not yet exhibiting symptoms, active case finding within the model is represented by flows (the orange arrows in Figure 1) from a broader set of compartments of undiagnosed exposed and infective individuals to the corresponding next stage diagnosed compartments of the model. It is to be noted that because of the multi-day time lag commonly associated with test results in the province, for both passive and active case finding, we let the flows of undiagnosed infectives proceed to the next stage diagnosed compartments instead of the directly corresponding diagnosed stages; thus, for example, those diagnosed from stage $I_{YU}$ flow into the next stage compartments of $I_{YND}$, rather than into $I_{YD}$.

The daily flow of cases being diagnosed by passive testing, but not leading to hospitalization, is mainly governed according to the endogenous model calculations $I_{YU}\left(f_{Y}\right)/t_{I_{Y}}$, where $f_{Y}$ is the fraction of undiagnosed symptomatic infected individuals with complications that do not require hospitalization during their course of infection, and $t_{I_{Y}}$ is the mean days to develop or avoid complications; this is bounded by the empirical data (denoted as $E_{m}$) of total test volume presenting other than due to hospitalization or international travel. $E_{m}$ can be calculated by the difference between the daily total test volume (denoted as $V_{t}$) and the three-way sum of daily admitted COVID-19 patients to ICU and non-ICU hospitalizations (denoted as $V_{HICU}$ and $V_{HNICU}$, respectively) and new likely exogenous cases (denoted as $E_{x_{D}}$). This difference reflects the known use of tests for hospitalization and the fact that out-of-province cases were carefully estimated for the opening weeks of the pandemic, and each required tests.

The model characterization of daily diagnosed cases identified specifically by active case finding—conducted via activities such as contact tracing, screening, and drive-through testing—is designed to capture the fact that in such forms of case finding, testing tends to drive the count of individuals diagnosed and identifies infected individuals at all stages of the natural history of infection. To represent the fact that test count drives the count of cases diagnosed with an efficiency limited by the number of infected individuals, we made use of a previously published testing model [28]. Within this model, the count of infectives identified by testing is characterized as $I_{U}\beta_{T}\left(1-e^{-\alpha \frac{V}{I_{U}}}\right)$, where *V* is the total test volume, $I_{U}$ is the total count of undiagnosed infectives, α is a measure of test efficiency, and $\beta_{T}\in \left[0,1\right]$ represents an upper limit on the fraction of infectives that could be identified via active case finding. In this test model, the term $\beta_{T}\left(1-e^{-\alpha \frac{V}{I_{U}}}\right)$ characterizes the fraction of all infectives that are diagnosed. Reflecting the fact that active case-finding efforts are incomplete in their reach, $\beta_{T}$ represents the fraction of infectives that would be diagnosed via active-case finding if the total test volumes *V* were to be arbitrarily large (i.e., the asymptomatic fraction of infected individuals who would be located as the ratio of test volume to infections approaches infinity); given the broad reach of contact tracing within the province, this work treated $\beta_{T}$ as 1. α is a measure of testing efficiency. When $\beta_{T}$ is 1 (as it is here), for a small active test volume *V*, this can be seen roughly corresponding to the product of the test positivity rate and test specificity: For every test performed, α infectives will on average be discovered. The saturating exponential term ($1-e^{-\alpha \frac{V}{I_{U}}}$) assumes that as the volume of tests performed for active case finding rises, a greater number of tests is needed to find a given infective. Thus, while more tests will identify additional infectives, doubling the count of tests performed will not double the count of infectives identified. By employing this test model to calculate the cases diagnosed by active case finding in this project and recognizing the priority placed on presentation-driven tests that drive passive case finding, the model assumes that the total volume of tests performed for active finding is given by the difference between the total testing volume ($E_{m}$) and the volume of tests performed for passive case finding $\left(\min\left(\frac{I_{YU}f_{Y}}{t_{I_{Y}}},E_{m}\right)\right)$, and thus, $V_{active}=E_{m}-min\left(\frac{I_{YU}f_{Y}}{t_{I_{Y}}},E_{m}\right)$. At any time, the total count of undiagnosed infectives can be calculated by summing all of the undiagnosed compartments, which is $I_{U}=E_{U}+I_{AU}+I_{A2U}+I_{A3U}+I_{YU}+I_{YNU}$. Thus, the model gives the diagnosed cases found by active testing as $V_{p}=I_{U}\beta_{T}\left(1-e^{-\alpha \frac{V_{\mathrm{active}}}{I_{U}}}\right)$. And the daily count of diagnoses from active case finding for different compartments (e.g., $E_{U}$, $I_{AU}$, $I_{A2U}$, $I_{A3U}$, $I_{YU}$, and $I_{YNU}$)—depicted as orange arrows in Figure 1—is treated as simply being split proportionally according to the count of people in each undiagnosed infective compartment.

#### 2.1.2. Acute Care Utilization

Undiagnosed or diagnosed symptomatic individuals who develop severe or critical COVID-19 complications [29] at the time of transitioning from the early-stage symptomatic period (leaving $I_{YU}$ and $I_{YD}$) are presumed to present for care and enter into the hospitalization stocks either for acute but non-critical care (compartment $H_{NICU}$) or for critical care ($H_{ICU}$)—the purple flows in Figure 1. The fraction of all individuals progressing from diagnosed early- to diagnosed late-stage symptomatic state who are treated as not developing severe or critical COVID-19 complications is treated as $1-f_{H}$. The fraction of all individuals progressing from undiagnosed early- to diagnosed late-stage symptomatic state diagnosed by passive testing is $f_{Y}$. And the fraction of individuals progressing from undiagnosed early- to undiagnosed late-stage symptomatic state diagnosed by passive testing is $1-f_{H}-f_{Y}$. Of the fraction $f_{H}$ of such progressors requiring hospitalization, the fractions that transition to the ICU ($H_{ICU}$) and non-ICU ($H_{NICU}$) are given by the parameters $f_{ICU}$ and $1-f_{ICU}$, respectively. Individuals in both such hospitalization compartments are further subject to mortality, with deceased individuals transitioning to compartment *D* at the time of passing, as indicated by the grey flows in Figure 1. Given the overall COVID-19 case fatality rate for hospitalized patients requiring ICU care or not in need of such care (denoted by $\phi_{ICU}$ and $\phi_{NICU}$, respectively), the model characterizes the corresponding daily mortality rates as $-ln\left(1-\phi_{ICU}\right)/t_{ICU}$ and $-ln\left(1-\phi_{NICU}\right)/t_{NICU}$, where $t_{ICU}$ and $t_{NICU}$ are the mean durations of ICU-hospitalized and non-ICU-hospitalized patient stays before death, respectively. As a simplifying assumption and to lower the count of compartments required and the resulting size of the state space, the model does not seek to explicitly model continued hospital residence amongst some patients prior to or following ICU discharge.

#### 2.1.3. Exogenous/Endogenous Infections

The model considers infectives as originating from both endogenous sources (via infection through contact with other infectives in the modeled population) and exogenous sources (where infectives arrive in the population via out-of-province (and particularly international) arrivals), which are flows represented by the magenta arrows in Figure 1. This exogenous flow is driven by the empirical time series of daily travellers infected outside of the population and was of strong importance for accounting for patterns in the opening two to three months of the pandemic, on account of the importance of international arrivals in driving subsequent endogenous transmission. Endogenous infections are calculated by the transmission system of the model.

#### 2.1.4. Vaccination System

The model considers two levels of vaccination-induced protection for the population [30]. This characterization reflects the fact that Saskatchewan’s vaccination campaign employed only two-dose vaccines, namely, Pfizer/BioNTech BNT-162b2, Moderna mRNA-1273, and AstraZeneca ChAdOx1. With the BNT-162b2 vaccine being responsible for approximately 74.86% of all vaccines delivered within the province, and conscious of the adverse impact on model state space size and—by extension—machine-learning inference accuracy, we made the simplifying assumption of characterizing vaccinated individuals by two levels of vaccine protection, rather than with further levels and/or via stratification with respect to each vaccine product. Two flows of daily vaccinated cases from the susceptible (compartment *S*) to the first level of vaccination-induced protection (compartment $V_{1}$) and from the first-dose vaccinated to a higher level of protection (compartment $V_{2}$) (represented by green arrows in Figure 1) are driven by the empirical time series of daily receipt of first-dose vaccines and second-dose vaccines. Because of limited evidence concerning the duration of vaccine protection [30], this model currently assumes the vaccines confer permanent protection. Individuals with both one and two doses of vaccines remain subject to the risk of infection, with the relative risk of infection in each dose-count-specific compartment compared to an unvaccinated symptomatic being given by one minus an estimate of vaccine efficacy against infection with that dose count. The vaccine efficacy against infection of the vaccines used within Saskatchewan is reported based on clinical trial data [31] that differ from vaccine to vaccine, notably against different COVID-19 variants of concern (VoCs). Reflecting the mixed vaccination regime and the presence of multiple VoCs over the timeframe of the study, the vaccine efficacy against infection considered in this project for a single dose (denoted as $e_{1}$) and two doses (denoted as $e_{2}$) are 0.8 and 0.95, respectively, based on the vaccines used in Canada—Pfizer, Moderna, and Astra-Zeneca [31]. While COVID-19 vaccines routinely offer greater efficacy against hospitalization and mortality than against infection, motivated in part by the desire to avoid the adverse effects on model inference of enlarging the state space of the model and lacking ready empirical data on breakthrough infections at the time of formulation, the model treats breakthrough infection as placing an individual into the same pathways of infection as are used for an infected unvaccinated individual.

#### 2.1.5. Infectious Transmission System

The force of infection parameter λ characterizes the hazard rate of infection—the probability density with which a fully susceptible (e.g., a person in the stocks of *S*) is subject to infection from an infective and is governed by mass action principles [32]. The force of infection parameter λ is calculated by $c\beta p_{e}$, where *c* is the contact rate among the population per unit time, β is the probability of transmitting COVID-19 per discordant contact, and $p_{e}$ is the effective prevalence of infectives in the mixing community. The construct of the effective prevalence of infectives in the mixing community, $p_{e}$, is designed to take into account the mixing implications of the symptoms, diagnosis, and acute care status of infective individuals; we refer to a relative-mixing-level-adjusted size of a subpopulation as the “effective” size of that subpopulation. The effective prevalence of infectives in the mixing community $p_{e}$ is represented by the fraction of the effective infectives among the effective population in the community. We assume that undiagnosed oligosymptomatic individuals (in compartments $I_{AU}$, $I_{A2U}$, and $I_{A3U}$) have full social contacts and undiagnosed symptomatic individuals (in compartments $I_{YU}$ and $I_{YNU}$) exhibit a relative reduction in the level of social mixing, as given by the fraction $\rho_{U}$, measured relative to full social contacts (themselves as assumed to be associated with a relative mixing rate of 1), and non-hospitalized diagnosed patients (in compartments $I_{AD}$, $I_{A2D}$, $I_{A3D}$, $I_{YD}$, and $I_{YND}$) in the community have a similar proportional reduction in mixing, denoted $\rho_{D}$. Hospitalized patients are treated as not engaging in mixing and thus do not contribute to the size of the effective mixing populations and carry a relative mixing rate of 0. It is important to emphasize that such values represent *relative* mixing rate characterizations; secular changes in contact rate across the population (such as those that might be caused by public health orders) are characterized by another element of the formulation detailed below. There are three flows in the model reflecting the force of the infection process—the infection from stocks *S*, $V_{1}$, and $V_{2}$, which are associated with rose-coloured flows in Figure 1.

#### 2.1.6. Municipal Wastewater Surveillance Characterization

Municipal wastewater refers to sewage containing waste from households, workplaces, and other sources served by municipal infrastructure [33]. In a public health context, wastewater surveillance (WWS) describes the process of sampling and analyzing wastewater to monitor phenomena such as the prevalence of conditions, use of pharmaceuticals, and occurrence of viral outbreaks in communities [33]. Medema et al. [34] demonstrated a significant correlation between COVID-19 virus SARS-CoV-2 concentrations in wastewater and the prevalence of COVID-19. This finding suggested that wastewater surveillance of SARS-CoV-2 could offer a tool to monitor the trends of COVID-19 prevalence in cities. Moreover, wastewater surveillance offers a significant advantage, since the concentration of SARS-CoV-2 in the wastewater sampling is representative of the entire population served by the sewage shed, regardless of health status, propensity for care-seeking behaviour, or reported infection status [33]. Moreover, because of the high shedding levels seen in the early stages of infection by SARS-CoV-2, wastewater assays can often identify presymptomatic or oligosymptomatic populations.

This project involved the design, implementation, deployment, and routinized use of a particle-filtered compartmental model to estimate the epidemiological and health system state using time series including wastewater concentrations of SARS-CoV-2. Due to the dynamics of viral load, fecal shedding in a SARS-CoV-2-infected individual varies across natural histories of infection, such as between symptomatic/asymptomatic, and over stages of progression [35,36,37]. We made use of a weighted shedding model reflecting the fact that individuals in the early stages of infection shed at far higher rates than do those at later stages of infection. Hoffman et al. [37] estimated that the shedding profile modulates viral concentrations in faecal samples over time. Figure 1 of [37] illustrates a generally exponential decrease in the shedding virus load over time, with the exception of the very early stage, when the virus load is relatively lower. Based on this figure, we conducted a rough estimation of the shedding virus load: we posit that the virus load is highest during the presymptomatic stage, and approximated the weight at 0.5; during the exposed and the early symptomatic and cotemporal stages of oligosymptomatic stages, we assume a weight of 0.2; during the symptomatic stages without complications and cotemporal stages of the oligosymptomatic stage, the virus load continues to decrease, and we assumed a weight of 0.1. The estimated weights of viral concentration of different stages based on this research are shown in Table 1. In light of the weighted shedding profile for individuals and the larger shedding populations of interest, we assume a constant of proportionality γ that relates the (weighted) value of the shedding population to the daily concentration of SARS-CoV-2. The value of γ is estimated by the PMCMC model, which can be referred to [38]. Reflecting the fact that the focus of wastewater monitoring within Saskatchewan was on cities exhibiting separated storm-water and wastewater infrastructure marked by short (≤8 h) toilet-to-municipal wastewater treatment plant transit times and the use of autosampling from the primary inflow into the treatment plant, we treated the concentration of COVID-19 wastewater samples for a given city as indicative of the current—rather than the lagged—epidemiology for that city. Finally, this paper primarily provides a concise introduction to the virus-shedding model, as our group is in the planning stages of another paper that will provide a detailed explanation of how wastewater data can be incorporated into COVID-19 models.

#### 2.1.7. Model Parameters

Table 1 gives the value and units for constant parameters of the deterministic COVID-19 model; readers interested in further detail regarding the formulations involving these parameters are referred to Appendix A.

### 2.2. Calculation of Variables of Interest from the COVID-19 Model

Figure 1 shows the system of ODEs governing the behaviour of the deterministic COVID-19 model. As detailed in Section 2.3.1, the stochastic version of this model serves as the state space model for particle filtering. We detail here a set of derived quantities whose formulation is identical for both forms of the model.

A variety of COVID-19 outcomes of interest can be derived from the ODEs shown in Equation (Equation 12) in Appendix A, including those relevant for epidemiological and acute care decision-making. From the standpoint of public health planning and epidemiology, important quantities include a dynamic characterization of the effective reproductive number (denoted as $R_{t}$), the count of undiagnosed infectives in the community with time (denoted as $N_{U}$), and the force of infection (λ). Each of these quantities provides information important for understanding the evolution of the current pandemic situation and played a central role in the reporting undertaken from the model. Such measures are especially useful in indicating the evolution of the epidemiological situation, anticipating incipient outbreaks, assessing the performance of current intervention strategies, and informing decisions to be made in the near future, such as those involving relaxation or re-imposition of public health orders.

Some of the model-derived values are of foremost value in the sphere of projection, rather than in the historic time horizon. From the standpoint of acute care and surge planning, the model offers particular value by virtue of its capacity to project forward acute care demand, both in the form of new admissions for COVID-19 to the intensive care unit (ICU) and non-ICU hospital needs and, in terms of census counts for both of those levels of acute care services. Particularly when the stochastic version of the model is used with particle filtering, such information can aid in decisions involving the triggering of surge capacity.

#### 2.2.1. Calculation of the Evolving Effective Reproductive Number

The basic reproductive number (denoted as $R_{0}$) and effective reproductive number (denoted here as $R_{t}$) are widely used concepts in mathematical epidemiological models. The basic reproductive number ($R_{0}$) is the average number of secondary infections transmitted by a typical infective individual in a completely susceptible surrounding population [43]. While an understanding of this quantity is of great value, in the context of an evolving outbreak, with a population of evolving susceptibility, behavioural and public health measure-induced changes in the contact rate, and changing variant ecology, greater day-to-day attention typically rests on the effective reproductive number ($R_{t}$). $R_{t}$ is the average number of secondary infections transmitted by a typical infective individual in a population composed of both susceptible and non-susceptible persons and reflective of the current epidemiology, including mixing patterns and public health, institutional and personal protective practices at present, vaccine effectiveness, population turnover, and currently circulating variants. As a general rule, if $R_{t}\left(t\right)>1$, the count of infected individuals will increase over time; if $R_{t}\left(t\right)=1$, the count of infected patients will remain roughly constant; if $R_{t}\left(t\right)<1$, the number of individuals infected will decline over time.

The model detailed here uses two methods to calculate the effective reproductive number ($R_{t}$): a simplified original method and a method that takes into account the differential mixing rates between undiagnosed and diagnosed individuals and the case-finding process that leads individuals to transition from the former to the latter. Both methods played prominent roles in daily reporting using the model throughout different stages of the pandemic. The original method is based on an assumption that all infectives exhibit full—not reduced—mixing with the susceptibles throughout their full duration of infectivity (i.e., until recovery). Recalling that $e_{1}$ and $e_{2}$ represent the vaccine effectiveness given one or two administered doses, respectively, and that $\rho_{U}$ and $\rho_{D}$ denote the relative rates of mixing amongst symptomatic but undiagnosed individuals and diagnosed individuals, respectively, the original values of $R_{0}$ and $R_{t}\left(t\right)$ in this COVID-19 model are characterized as follows:(1)$$\begin{array}{cc} \; & R_{0}=C\beta \left(t_{I}+t_{I_{Y}}+t_{I_{YN}}\right)\\ \; & R_{t}\left(t\right)=R_{0}\cdot f_{Susc}\left(t\right)\\ \; & f_{Susc}\left(t\right)=\frac{S\left(t\right)+\left(1-e_{1}\right)V_{1}\left(t\right)+\left(1-e_{2}\right)V_{2}\left(t\right)}{N(t)}\\ \; & N\left(t\right)=\left(S+E_{U}+I_{AU}+I_{A2U}+I_{A3U}+R_{U}+V_{1}+V_{2}\right)+\rho_{U}\left(I_{YU}+I_{YNU}\right)+\\ \; & \;\;\rho_{D}\left(I_{AD}+I_{A2D}+I_{A3D}+I_{YD}+I_{YND}+R_{D}\right) \end{array}$$

However, in real-world scenarios (and in this model), infection spread is governed by other factors besides those captured in the equations above. Specifically, the degree of infection spread from an infective is affected by the relative mixing levels between undiagnosed symptomatics and diagnosed infectives. Whilst the characterization in Equation (Equation 1) considers those factors inasmuch as they affect the fraction of contacts that are made with susceptibles, it fails to consider them in terms of the behaviour of the infective individual over the course of their illness. Considering the effective duration infectives spend in different infected stages leads to a new formulation for each of the basic and effective reproductive numbers, denoted $R_{0}^{\prime }$ and $R_{t}^{\prime }$, respectively:(2)$$\begin{array}{cc} \; & R_{0}^{\prime }=C\beta t_{Effective}\\ \; & t_{Effective}=\frac{1}{I_{E}}[\left(I_{A}+\rho_{D}I_{AD}\right)\left(t_{I}+t_{I_{Y}}+t_{I_{YN}}\right)+\\ \; & \;\;\;\left(I_{A2}+\rho_{D}I_{A2D}+\rho_{u}I_{YU}+\rho_{D}I_{YUD}\right)\left(t_{I_{Y}}+t_{I_{YN}}\right)+\\ \; & \;\;\;\left(I_{A3}+\rho_{D}I_{A3D}+\rho_{u}I_{YD}+\rho_{D}I_{YND}\right)t_{I_{YN}}]\\ \; & I_{E}=\left(I_{A}+I_{A2}+I_{A3}\right)+\rho_{u}*\left(I_{YU}+I_{YN}\right)+\rho_{D}*\left(I_{AD}+I_{A2D}+I_{A3D}+I_{YUD}+I_{YND}\right)\\ \; & R_{t}^{\prime }\left(t\right)=R_{0}^{\prime }\cdot f_{Susc}\left(t\right) \end{array}$$

In this contribution, we employ the latter method, which considers the effective time of infectives to estimate and predict the effective reproductive number $R_{t}$.

#### 2.2.2. Count of Undiagnosed Infectives in the Community over Time

Given the underlying structure of the model, the count of undiagnosed infectives in the community $N_{U}\left(t\right)$ can be calculated by summing the count of undiagnosed persons in each infective compartment as follows:(3)$$N_{U}\left(t\right)=I_{AU}+I_{A2U}+I_{A3U}+I_{YU}+I_{YNU}$$

#### 2.2.3. Daily Effective Prevalence of Infectives in the Mixing Community

The point prevalence of COVID-19 is the proportion of individuals in a population who have COVID-19 at a specified point in time [44]. Thus, the equation of the standard prevalence is as follows:(4)$$\begin{array}{c} p_{st}=\frac{I_{AU}+I_{A2U}+I_{A3U}+I_{YU}+I_{YNU}+I_{AD}+I_{A2D}+I_{A3D}+I_{YD}+I_{YND}}{S+E_{U}+I_{AU}+I_{A2U}+I_{A3U}+V_{1}+V_{2}+I_{YU}+I_{YNU}+I_{AD}+I_{A2D}+I_{A3D}+I_{YD}+I_{YND}} \end{array}$$

In the model, we use the effective prevalence instead of the standard prevalence. The effective prevalence considers the weight of contact coefficients of the undiagnosed infectives ($\rho_{U}$) and the weight of contact coefficients of the diagnosed infectives ($\rho_{D}$). Thus, the daily effective prevalence of infectives in the mixing community can be calculated by the fraction of the effective infectives in the total effective population in the community. The formulation is as follows:(5)$$\begin{array}{c} p_{t}=\frac{\left(I_{AU}+I_{A2U}+I_{A3U}\right)+\rho_{U}\left(I_{YU}+I_{YNU}\right)+\rho_{D}\left(I_{AD}+I_{A2D}+I_{A3D}+I_{YD}+I_{YND}\right)}{\left(S+E_{U}+I_{AU}+I_{A2U}+I_{A3U}+V_{1}+V_{2}\right)+\rho_{U}\left(I_{YU}+I_{YNU}\right)+\rho_{D}\left(I_{AD}+I_{A2D}+I_{A3D}+I_{YD}+I_{YND}\right)} \end{array}$$

#### 2.2.4. Force of Infection

Section 2.1.5 introduced the model’s use of the force of infection (λ). This quantity can be calculated as the product of what we term the transmission rate—itself the product of the contact rate and probability of transmission per discordant contact—and the fraction of the mixing population that is infectious:(6)$$\begin{array}{cc} \; & \lambda =c\beta p_{t} \end{array}$$
where $p_{t}$ is the daily effective prevalence of infectives, as characterized by Equation (Equation 5).

#### 2.2.5. Cumulative Prevalence of Infections

Period prevalence is the proportion of individuals in a population who have had COVID-19 over a specified period of time [44]. Thus, the cumulative prevalence of COVID-19 infections can be calculated by the fraction of the initial population who have ever been infected by COVID-19. The formulation of the cumulative prevalence of infections at time *T* is as follows:(7)$$p_{c}=\frac{\int_{0}^{T}\lambda \left[S+\left(1-e_{1}\right)V_{1}+\left(1-e_{2}\right)V_{2}\right]\;dt}{N_{0}}$$

#### 2.2.6. New Hospital Admissions and Census Count for Non-ICU and ICU Needs

A key motivator for the construction of the COVID-19 model characterized in this project is to estimate and predict acute care demand and capacity utilization. This includes considering hospital admissions—including ICU admission and non-ICU admission cases—and the daily number (census) of ICU and non-ICU hospital patients. The daily hospitalized census for ICU and non-ICU at any given time is simply characterized by the values of the compartments $H_{ICU}$ and $H_{NICU}$, respectively. Recalling that the time unit of the model is days, the per-day rate (daily count) of new admissions of ICU patients is given by the sum of two flows into the $H_{ICU}$ compartment, representing the development of critical symptoms by both previously diagnosed (IYD) and (separately) previously undiagnosed (IYU) symptomatic infectives. Similarly, the daily new hospital admissions of patients not requiring ICU care is the sum of two flows into the $H_{NICU}$ compartment, representing the development of severe symptoms by both previously diagnosed (IYD) and undiagnosed (IYU) individuals. Thus, the daily new admissions of ICU and non-ICU patients are as follows:(8)$$\begin{array}{ccc} \; & dH_{ICU}=\frac{\left(I_{YU}+I_{YD}\right)f_{H}f_{HICU}}{t_{I_{Y}}}\; & dH_{NICU}=\frac{\left(I_{YU}+I_{YD}\right)f_{H}\left(1-f_{HICU}\right)}{t_{I_{Y}}} \end{array}$$

### 2.3. SMC Algorithm Incorporation of the Stochastic COVID-19 Model

The prominent sequential Monte Carlo (SMC) method of particle filtering is a contemporary state inference and identification methodology that supports filtering of general non-Gaussian and non-linear state space models in light of time series of empirical observations [3,5,8]. This approach estimates, via sampling, the time-evolving internal state of a dynamic system (here, the COVID-19 model) in which random perturbations are present and where information about the state is obtained via noisy measurements made at each observation time. The state space model characterizes the processes governing the time evolution of the internal state of the system with stochastics consisting of random perturbations. The state of the state space model is assumed in general to be latent and unobservable. Information concerning the latent state is obtained periodically via a noisy observation vector. The particle filtering method can be viewed as undertaking a “survival of the fittest” of varying hypotheses as to the current location of the system in state space, with each such hypothesis being represented by a particle, the fitness of which is determined by the consistency between what is observed empirically at each observation time point and what would be expected given the state of the particle (the hypothesized state) at that time point. Interested readers are referred to a more detailed treatment in [5,8,9].

#### 2.3.1. State Space Model

The state space model depicts the processes governing the time evolution of the state—both latent and observable—of a noisy system. In this paper, the state space model consists of a stochastically embellished variant of the deterministic COVID-19 model depicted in Figure 1 and whose equations are given in Equation (Equation 12) of Appendix A. Reflecting the fact that effective use of particle filtering requires an underlying state equation model exhibiting stochastic variability, we characterize here an extension of the deterministic model that incorporates random perturbations in dynamic processes—including several stochastically evolving parameters—so as to reflect stochastic time evolution in the external world. The extended stochastic model introduced in Figure 2 then serves as the basis for an accompanying particle filter.

The state vector of the particle filtering model is given by:$$\begin{array}{c} {}[S,E_{U},I_{AU},I_{AD},I_{A2U},I_{A2D},I_{A3U},I_{A3D},I_{AU},I_{AD},I_{YU},I_{YD},I_{YNU},I_{YND},H_{ICU},H_{NICU},\\ R_{U},R_{D},D,logit\left(C_{\beta }\right),logit\left(\alpha \right),logit\left(f_{H}\right),logit\left(f_{Y}\right){]}^{T}. \end{array}$$

##### Dynamic Processes

We consider stochastic processes to characterize the arrival of undiagnosed travel-imported symptomatic cases, contact and care-seeing behaviour, and test positivity rates associated with active screening. Moreover, Poisson processes are used to reflect the stochastics associated with the occurrence of a small number of cases over a small unit of time—denoted as Δt (carrying the value of 0.001 days in the COVID-19 model) [3,5]. The stochastic process characterizing undiagnosed travel-based importation of symptomatic infectives is given by $\frac{Poisson\left(Ex_{D}\Delta t\frac{1-f_{S}}{f_{S}}\right)}{\Delta t}$.

##### Dynamic Parameters

There are a set of quantities that might commonly be regarded as parameters, but whose values evolved in notable ways over the course of the COVID-19 pandemic, particularly with the evolution of human behaviour and variant ecology, due to changes in active case-finding efforts and the arrival of the pathogen in vulnerable demographics and communities. Such quantities are termed “dynamic parameters” herein. The dynamic parameters of the deterministic COVID-19 model are listed in Table 2.

#### 2.3.2. Likelihood Function

Our formulation of the overall likelihood function and sub-likelihood functions for this work drew inspiration from our past success in employing negative binomial-based likelihood functions in a diverse set of particle filtering applications in communicable disease [3,4,5,6] and by others in MCMC-based approaches for H1N1 influenza [45]. Moreover, for simplicity and in line with formulations used successfully in multiple of our past contributions [3,4,6], the current model characterized the overall likelihood function for the particle filtering model as the product of sub-likelihood functions, each considering a distinct subset of the empirical datasets employed to ground the model:(9)$$\begin{array}{cc} L=\; & L_{NewReportedEndogenousCases}\times L_{CumulativeReportedEndogenousCases}\\ \; & \times L_{CumulativeICUAdmissions}\times L_{CumulativeNICUAdmissions}\\ \; & \times L_{ICUCensus}\times L_{NICUCensus}\\ \; & \times L_{CumulativeDeaths}\times L_{ViralConcentrationsinwastewater} \end{array}$$

Each sub-likelihood function is characterized by one of two distinct parametric statistical distributions—a negative binomial distribution or gamma distribution. Such sub-likelihood functions characterize the likelihood of observing the empirical datum, given an underlying model state specified by the particle state. Those two forms of sub-likelihood functions are introduced as follows:The value of each sub-likelihood function based on a negative binomial distribution is given as follows:
(10)LNegativeBinomial=y+r−1r−1pr(1−p)y
where *y* is the observed datum, *x* is the model value corresponding to that datum (integer rounded), *r* is the dispersion parameter associated with the negative binomial distribution, and $p=\frac{x}{x+r}$. In this project, the value of the dispersion parameter *r* was chosen to be 5.The value of the sub-likelihood function based on a gamma distribution is given as follows:
(11)$$L_{Gamma}=\frac{\beta^{\alpha }y^{\left(\alpha -1\right)}e^{-\beta y}}{\int_{0}^{\infty }z^{\alpha -1}e^{-z}dz}$$
where *y* is the observed datum, *x* is the model value corresponding to that datum, *k* is the shape parameter, $\alpha =\frac{x}{k-1}$, and $\beta =\frac{k}{x}$. Such likelihood functions within this project assumed a value of k=5.

It is important to note that while the likelihood function employed here is designed to be used with each of the types of data shown in Table 3, the likelihood formulation is moreover designed to be robust in the context of missing data for several of those types of data. Data that can be accommodated as missing include hospitalized admission data—ICU and non-ICU, hospitalized census data—ICU and non-ICU, and viral concentration in wastewater data. When a datum is not available for these types of observations, the corresponding sub-likelihood function will be treated as holding a value of unity (1.0). Thus, given missing data of this sort, the overall likelihood function will still carry the value of the product of the sub-likelihood functions for which data are available. Moreover, the model’s form can be readily extended to accommodate the handling of delayed data. Our other work [4] also demonstrates that even lower-quality data can, at times, contribute significantly to the model’s predictive accuracy.

### 2.4. Data Sources

Since June 2020 and through early 2022—and with prior episodic use—this model has served in a production capacity for the whole of Saskatchewan, and for varying periods for particular regions, municipalities, and small-area geographies within Saskatchewan. For the period October 2020–October 2021, via a contract with the Public Health Agency of Canada (PHAC), it was further used for reporting and projections multiple times a week for all provinces of Canada. Beyond that, via a contract for reporting to the First Nations and Inuit Health Branch of Health Canada (FNIHB), the model was used in the period November 2020–March 2022 for biweekly reporting and projections for First Nations Reserves in six Canadian provinces. Most such uses have exercised subsets of the likelihood functions considered, with hospital census data and wastewater data being restricted to subsets of jurisdictions.

For a given jurisdiction, empirical datasets are fed into the particle filtering model to estimate and predict the evolution of the epidemiological and acute care state of that jurisdiction. The empirical datasets employed in the model can be divided into two categories: a set incorporated in the likelihood function for training the particle filtering model and another that serves as an exogenous input to the differential equation model.

The following empirical datasets were considered in the likelihood function:Daily count of new reported incident confirmed or suspected cases.Cumulative reported incident confirmed or suspected cases from the inception of the pandemic.Cumulative reported deaths from COVID-19.Daily count of COVID-19 patients admitted into the ICU.Daily COVID-19 patients admitted into hospital for non-ICU care.Daily midnight census (count) of COVID-19 patients in the ICU.Daily midnight census of COVID-19 patients in the hospital for non-ICU care.Weekly average virus SARS-CoV-2 concentration in wastewater.

The following empirical datasets were incorporated as exogenous inputs directly into the dynamic model:Daily new likely exogenous cases, which represent arrivals into the jurisdiction believed to be infected while outside the jurisdiction, with an emphasis on international travel.Daily count of persons undergoing PCR (nasopharyngeal swab)-based testing.Daily count of COVID-19 patients admitted into the ICU.Daily count of COVID-19 patients admitted into hospital for non-ICU care.Daily count of persons who received their first vaccination dose.Daily count of persons who received their second vaccinate dose.

It is important to note that the empirical datasets of “Daily count of COVID-19 patients admitted into the ICU” and “Daily COVID-19 patients admitted into hospital for non-ICU care” are used in both the likelihood function and in driving the model directly.

### 2.5. Characterizing Model Fidelity to Empirical Data

A key driver for the evolution of the particle filtering approach applied here was the ongoing critical assessment of the fidelity between outputs from the particle-filtered model and the above empirical data sources. As a primary metric for assessing such fidelity in this project, we employed a discrepancy function. The discrepancy between the particle filtering model results and each empirical dataset specified here is the mean of the normalized RMSE (root mean square error) across the whole time frame of the model incorporating the empirical data; as such, smaller discrepancies are considered favourable. To accommodate the different scales of multiple empirical datasets, we employ the normalized RMSE to measure the difference between the model-estimated/predicted values and the observed data of each empirical dataset on each day having observed data. The mathematical formulation for the normalized RMSE is specified in Appendix D.

## 3. Results

This section characterizes the COVID-19 particle filtering model results and (empirical data availability permitting) associated discrepancies for both day-to-day estimates of the epidemiological state and projection of quantities such as the future daily infected cases, force of infection, and ICU and non-ICU admissions and census.

### 3.1. Particle Filtering Model Results with Incorporating Empirical Datasets

Although the particle filtering model characterized here at various intervals provided reporting for 17 different jurisdictions, for the sake of simplicity, we focus here on the results for a jurisdiction offering wastewater data and served by one of the longest spans of data—Saskatoon, Saskatchewan. In the application examined here—which is emblematic of simulations conducted on this jurisdiction over long periods of time—the COVID-19 particle filtering model takes in daily incoming empirical data to produce daily reporting. The model runs start on 22 February 2020, when the empirical data became available from the appropriate public health agency (here, the Saskatchewan Health Authority); testing for COVID-19 began on 25 February 2020, and the first reported infected cases occurred on 11 March 2020. The simulation here proceeds to 31 July 2021, prior to the widespread appearance of the Delta variant of concern. Particle filtering was conducted with a particle count of 150,000.

Table 4 presents the mean discrepancy, with five runs of the model. For comparison in scale, the table further provides the 95% confidence interval of each empirical dataset incorporated in the likelihood function to ground the model. As a reminder, the lower the discrepancy, the better the results sampled from the particle filtered model reproduce the empirical dataset.

Figure 3, Figure 4, Figure 5, Figure 6, Figure 7, Figure 8, Figure 9 and Figure 10 show the particle filtering COVID-19 model’s (the results of the minimum discrepancy among those five runs) estimated results compared with empirical data. The comparison between the model results and empirical data indicates that the particle filtering COVID-19 model can estimate the daily COVID-19 transmission and hospitalization status. As an important caveat, for data confidentiality reasons, precise empirical data are only provided here for empirical data publicly available through the Saskatchewan Health Authority COVID-19 dashboard [46]. For depiction of the two types of data not publicly available (ICU and non-ICU hospital admissions) in those figures, we ensure data confidentiality by showing synthetic data in the figure instead of actual data. Specifically, for each of ICU and (separately) non-ICU hospital admissions, the data shown in the figures for a given day are Poisson-distributed pseudo-empirical data. That is, for day *t* with an actual count of $n_{t}$ hospital admissions, the synthetic datapoint is drawn from $poisson(\max\left(n_{t},0.05\right))$.

### 3.2. Estimation of Latent Dynamic Variables

Through ongoing incorporation of the empirical datasets to ground the COVID-19 dynamic model, the particle filtering process estimates the latent states and dynamic variables to inform the COVID-19 transmission. Figure 11 shows the model-estimated daily effective reproductive number (the method and underlying mathematical formulation can be found in Section 2.2.1). Figure 12 shows the model-estimated daily undiagnosed infectives (with details on formulation found in Section 2.2.2). Figure 13 shows the model-estimated force of infection (λ) (with details on formulation found in Section 2.2.4). Figure 14 shows the model-estimated daily effective prevalence of infectives in the mixing community (with details on formulation found in Section 2.2.3). And Figure 15 shows the cumulative prevalence of infections for each day (with details on formulation found in Section 2.2.5). Readers interested in the estimated latent state of the COVID-19 particle filtering model are referred to Appendix D.

### 3.3. Projection Results

While the COVID-19 particle filtering model offers strong performance in monitoring and estimating COVID-19 transmission, throughout its use across jurisdictions, such estimates of current epidemiological state have routinely been accompanied by 14-day projections of COVID-19 transmission and hospitalization.

To assess the predictive capacity of the COVID-19 particle filtering model herein, we employed it to perform 1-day, 7-day and 14-day predictions for each day starting from day 100 in Saskatoon and continuing for the remainder of the time horizon considered here. Within the projection period (e.g., 7 days), no further particle filtering is performed, the model is simply run forward without any incorporation of the observed data. Mirroring the process seen in de facto use of the model, such a projection is performed from each successive day. It is essential to recognize that while the projection itself does not incorporate any new data, *between each day on which the projection is launched, new data arrive and are incorporated by the particle filtering*. The updated estimate of the latent state of the system afforded by this incorporation of the new data by particle filtering allows the next projection (looking into the “future”, as obtained from the standpoint of that day) to be made on the basis of the refined and updated understanding of the current epidemiological context.

Table 5 shows the predictive discrepancy between the model-predicted results and the empirical data. By comparing the average discrepancy of the three projection runs—1 day, 7 days, and 14 days ahead—we can see that the relative accuracy of the projections decreases with longer prediction timeframes. It bears emphasis that while discrepancies are computed here to compare the model results against empirical data, during the projection timeframes launched from each day, the empirical data are only used for comparison with the model-projected results. However, as noted above, new data are taken into account before undertaking each successive projection.

Figure 16, Figure 17, Figure 18, Figure 19, Figure 20 and Figure 21 depict a comparison between the model-predicted results and the empirical datasets (or, for confidentiality of hospital admissions, the empirically inspired synthetic data noted above). To understand the results, it is to be emphasized that for every day, we perform three predictive runs (1-day, 7-day, and 14-day). When shown in boxplots in the figures, the values for (one-)day-ahead predictions of the model will be shown directly in comparison with the corresponding day-ahead empirical data. In contrast, for the 7-day projection results, for each day the figures visually compare the average model-predicted value over that 7-day interval with the corresponding average of the empirical data over that same 7-day interval. As above, it is to be emphasized that within each such projection from a given day, no particle filtering is occurring, and the empirical data are only compared with the model results, not incorporated into the model. As noted above, as would and did occur in day-to-day practice of a deployed system such as this, with the passage of each successive day, new data are incorporated by the particle filtering mechanism to update the estimate of the system state, allowing the next projection to be made on the basis of that updated state estimate.

Those figures show that the preponderance of observed data (blue points in the diagrams) fall within the 50% inter-quartile range of the boxplot, demonstrating relatively accurate model predictions for up to 14 days in advance to inform public health agencies and governments, as in estimates informed by data up to the point of projection. For example, the prediction of daily new reported cases can produce a picture of the trends in future transmission, as informed by the system state estimated by the particle filtering. In a similar manner, the prediction of daily hospitalized ICU, non-ICU admission, and census patients can inform public health agencies’ mobilization of surge capacity, anticipation of capacity utilization, and service demands, and more broadly, support judicious allocation of hospital resources over the multi-week timeframe.

Finally, readers interested in the sensitivity analysis of the model parameters, please refer to a previous study [47] by our group that evaluated the particle filtering infectious disease models’ prediction accuracy with the influences of the frequent observations of empirical data, the parameters of the negative binomial dispersion parameters, and rates with which the contact rate could evolve.

### 3.4. Intervention Results

In the previous sections, we showed that the particle filtering algorithm can estimate the state space of the COVID-19 model. Beyond supporting the projection methods examined in the previous session, the capacity to perform such state estimation also confers benefits for conducting simulations of tradeoffs between intervention strategies, despite their counterfactual character. As for projection scenarios discussed in the previous section, during the time horizon of a given intervention run (e.g., 14 days), the particle filtering is disabled, and the dynamic model is run forward with no empirical data being incorporated. But, as for the projection methods above, with each successive day of operation, particle filtering incorporates a new day’s worth of data. The accuracy of the intervention runs conducted forward into the future from a given day following the particle filter update from the previous day benefits from the updated state estimate made possible by particle filtering’s incorporation of a new day’s worth of data.

In this section, we show two intervention experiments to simulate stylized public health intervention policies. The stylized intervention strategies are characterized abstractly for demonstration purposes, but are emblematic of the sort of more textured interventions examined during use of the model and provide a flavour of what could be achieved with other interventions. In each case, the scenarios are run forward for 14 days from each successive day with the intervention mechanisms in place. Such a scenario run undertaken from a given day depicts the posited result of undertaking the associated intervention starting on that day.

For a given such day, a single boxplot for that day depicts, on the basis of the state estimate as of that particular day as updated by particle filtering for that day, the average outcome over the 14-day intervention time horizon. It bears emphasis that, in each case, the intervention scenario is counter-factual—the intervention is not put into effect from day to day; rather, each 14-day intervention scenario projects what the impact of such an intervention would be, were it to be undertaken starting on the current day.

The two interventions considered here are predominantly focused on actions undertaken during two outbreak waves:The first stylized intervention exhibited here focuses on elevating hygiene-oriented personal protective measures, such as might be exemplified by a regional mask mandate. For simplicity, the examination here characterizes such interventions as multiplying the effective contact rate by a coefficient in the range (0,1). Figure 22 depicts the results of such a counterfactual scenario occurring, focused on the first outbreak wave in Saskatoon. For simplicity, this scenario posits an aggressive hygiene-enhancing intervention that reduces the contact rate by 50% specifically for the window between day 220 and day 310 (inclusive).In the second intervention type, we examine the outcomes from a stylized outbreak-response immunization campaign elevating vaccination rates for the 14-day defined period. This effect is achieved by using a coefficient to increase the effective vaccination rate in the model over that timeframe. As an example, Figure 23 shows the results of elevating the effective vaccination rate by 50% during the third outbreak wave in Saskatoon, with those elevated rates being in place from day 390 to day 510, inclusive.

The baseline comparator for the intervention runs—no intervention policies performed—can be found as the normal projection runs in Figure 16, Figure 17, Figure 18, Figure 19, Figure 20 and Figure 21.

## 4. Discussion and Limitations

As demonstrated in its use to support reporting for 17 jurisdictions across Canada for a period of a year or more, the COVID-19 particle filtering model can monitor COVID-19 transmission and hospitalization, estimate the daily latent states and important dynamic variables, and predict future daily transmission and hospitalization status over a multi-week timeframe, all in light of daily updates to the estimated system state. When running the model daily, the daily estimates for COVID-19 transmission, hospitalization status, projections, and intervention results reflect the latest sets of empirical data—including both health system and wastewater data—to provide a current understanding to inform public health and healthcare system decision-making.

This work suffers from a number of limitations. A key one concerns changes in variant ecology. Reflective of the high amounts of transmission experienced globally, the virus SARS-CoV-2 causing COVID-19 has exhibited marked evolution. For most of the period for which data are considered in this paper (early 2020 to the end of July 2021), the wild type of SARS-CoV-2 was the uniform lineage in place in Canada, with the Alpha variant [26] appearing in Canada in the final week of 2021 (and in Saskatchewan by February 2021), followed by Beta, Gamma, and Delta. With respect to the jurisdictions considered in our example runs here (Saskatoon), the highly distinctive Delta variant [26] became the dominant variant of SARS-CoV-2 following July 2021 driving the next wave of outbreaks. Our COVID-19 particle filtering model is capable of simulating changes in the virus ecology by adjusting the characteristic parameter values (e.g., to reflect virulence, transmissivity, fraction of cases that are symptomatic, or vaccine effectiveness). For example, to change from simulating the Alpha to the Delta variant, we increased the maximum value of the “transmission contact rate” (denoted $c_{\beta }$ in this paper) and decreased both of the two doses’ vaccine efficacy (denoted $e_{1}$ and $e_{2}$). However, the dynamic model assumes the presence of a single variant at a time and is not suitable for characterizing processes requiring representation of multi-variant ecologies, such as those involving competition between multiple lineages. It is also not well-suited for capturing variant cross-reactivity with respect to immunological protection.

Although the structure of the COVID-19 model has demonstrated effectiveness in simulating COVID-19 transmission and hospitalization across diverse jurisdictions from the beginning of the first infected individual occurrence until the end of 2021, there are a number of key shortcomings in the existing structure of the model. Likely the single most important such limitation relates to the failure of the model to adequately characterize the differential impact of vaccination on protection from infection vs. protection from severe disease and death. The model’s existing characterization of COVID-19 vaccination characterizes its impact of vaccination only as mediated by an impact on transmission. While individuals in the model can be infected regardless of vaccination status, once a breakthrough infection of a vaccinated individual occurs, the model lacks existing mechanisms for retaining information on that individual’s vaccination status. As a result, conditional on infection, the model grossly unrealistically characterizes a vaccinated individual as having an identical risk of hospitalization, ICU admission, and death to a non-vaccinated individual. While the model parameters associated with such outcomes can be modified to reflect a high prevalence of vaccination uptake, the model urgently needs a means of characterizing different types of protection conferred by vaccines. Such a representation is particularly urgent in light of the need to capture the evolution of variant ecology emphasized above. However, it is notable that the model presented in this manuscript possesses the capability to capture, to a certain extent, the protective effects of vaccination against severe disease and death. A dynamic variable, denoted the “daily fraction of symptomatic individuals who proceed on to require hospitalization” (represented as $f_{H}$), is defined in the model. This variable undergoes dynamic changes on a daily basis (with the time unit of the model being “day”) and is estimated using the particle filtering algorithm incorporating empirical datasets. Beyond this foundational modification, the model requires the capacity to represent the impact of successive booster vaccines.

Beyond the key change required for the characterization of vaccine-induced protection, the model depicted in this paper exhibits a need to adapt to the updated epidemiological context and evolved understanding of SARS-CoV-2. Most important is the need to take into account the extensively evidenced phenomenon of waning of both natural immunity (acquired from exposure to the disease through infection) and vaccine-induced immunity.

The COVID-19 model structure characterized here is only applied to an aggregate population. Important gains in insight can be secured by the incorporation of key elements of heterogeneity via stratification. Given the marked differences in risk of severe disease and hospitalization, vaccine uptake, assortative mixing, and risk behaviour, stratification by age group is a key priority. Particularly in light of the pronounced rural–urban disparities in vaccination and risk behaviour and opportunities for incorporation of data drawn from SARS-CoV-2 wastewater concentration assays across varying municipalities, stratification by multiple regions could also confer notable benefits.

Most of the needs covered in this section have subsequently been successfully incorporated into newer versions of the particle filtered dynamic model than those presented here, but coverage of this expanded model and particle filtering framework lies outside this presentation. For instance, model structure modifications were made to the vaccination parts, such as adding the third booster vaccination dose and incorporating waning immunity, during simulation of the Omicron variants.

Also left for separate coverage is our refinement and expansion of the model covered here into a particle Markov chain Monte Carlo model offering additional capabilities and sophistication in sampling of static parameters [38], closer examination and evaluation of the incorporation, and support for drawing insight from wastewater data. Details of the extensive and articulated distributed computation framework used to provide nearly fully automated day-to-day running and reporting of the model results across diverse jurisdictions and data sources at scale are also covered elsewhere.

Last but not least, a notable limitation of the particle filtering model lies in the potential impact of the size of the model’s state space on its results. While extending an aggregated population compartmental model to include multiple stratifications is both insightful and straightforward, it is crucial to recognize that the model’s state space will also significantly expand. Managing a larger-state-space particle filtering model often poses challenges in testing and tuning. Additionally, increased computational resources, such as a larger number of particles, are also a challenge due to a larger state space model. To effectively ground and estimate the state space model, more empirical datasets become necessary. Therefore, multi-dimensional stratified population particle filtering models (for instance, those stratified by multiple age groups or geographical regions, incorporating multiple simulated virus variants in the entire model, or accounting for individuals’ vaccination history during post-infection periods) do not consistently ensure improved results compared to aggregate population models.

## 5. Conclusions

This study characterizes the design and multi-year deployment of a production-quality particle filter model that played a central role in informing public health decision-making starting in the opening months of the pandemic. By cross-leveraging particle filtering, dynamic (transmission) modeling, and diverse health system and wastewater data sources, the system presented here and close variants offered important initial findings by April 2020 and served to deliver daily updated COVID-19 situational analyses and short-term forecasts for Saskatchewan for the period of June 2020 through December 2021, multiple times a week for each Canadian province for the Public Health Agency of Canada until November 2021. and weekly to First Nations across six Canadian provinces via FNIHB through March 2022.

Particle-filtered dynamic models confer strong benefits by virtue of their ability to incorporate diverse incoming empirical data streams—here including both regularly reported health system data and episodically sampled wastewater data—to perform day-to-day probabilistic estimation and reporting of latent epidemiological and health system quantities of interest. Quantities routinely reported from the model described here include—but are not limited to—COVID-19 cases, testing volumes, hospitalization admissions and census, deaths, force of infection, undiagnosed individuals, and other factors. This further includes a more sophisticated estimate of the effective reproductive number, taking into account incomplete reporting, asymptomatic transmission, diagnosis, isolation, and other considerations. Beyond supporting updated estimation of such quantities and other elements of the system state whenever new data arrive, our extensively deployed particle-filtered framework uses each new system state estimate as the basis for probabilistically projecting forward the evolution of epidemiology and acute-care demand, which can readily support the triggering of surge capacity mobilization, motivate the institution of public health measures, or prepare for higher health capacity utilization. Similar methods can and were used to support the reporting of results from prospective counterfactual intervention scenarios, with each undertaken in light of the latest empirical observations.

As demonstrated by its widespread adoption for continually regrounded reporting and scenario analysis for diverse Canadian jurisdictions, the sequential Monte Carlo approach of particle filtering offers a compelling tool for evidence-based public health decision-making. The capacity of particle filtering to keep transmission models and the resulting probabilistic state estimates and scenario projections continually updated with the latest data offers compelling advantages over earlier generations of techniques such as the extended Kalman filter, and the computational demands of this technique are well-balanced with the velocity of contemporary data streams of relevance. Systems employing particle filtering offer strong advantages well-matched to the urgent need for public health surveillance and decision-making in the coming years.

## Figures and Tables

**Figure 1 ijerph-21-00193-f001:**
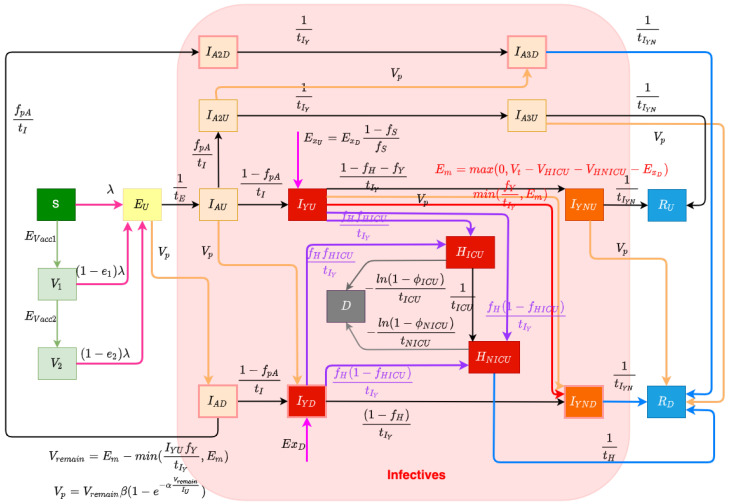
Transmission model structure.

**Figure 2 ijerph-21-00193-f002:**
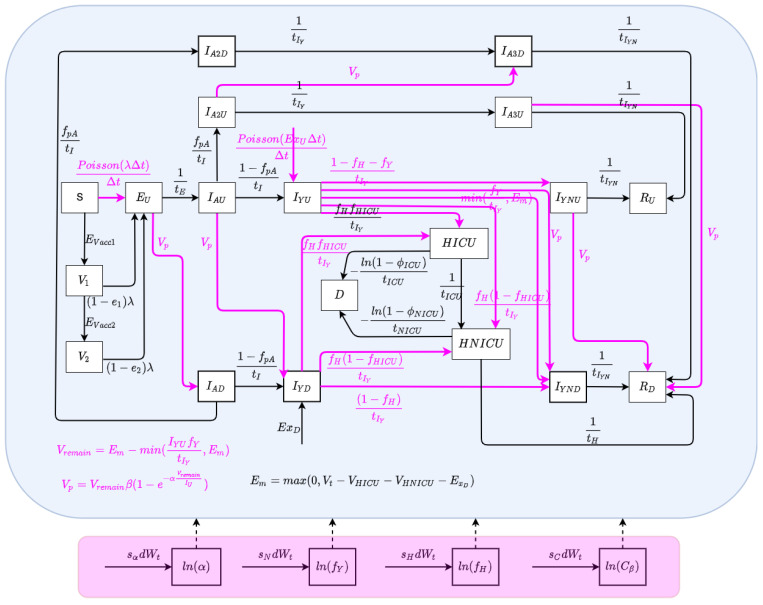
The model structure of the stochastic particle filtering model.

**Figure 3 ijerph-21-00193-f003:**
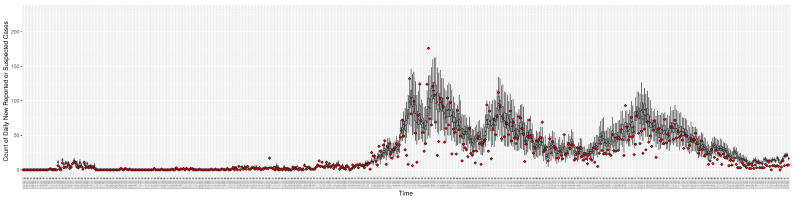
Daily new reported confirmed or suspected infective cases between particle filtering model results (boxplot) and empirical data (superimposed red scatterplot).

**Figure 4 ijerph-21-00193-f004:**
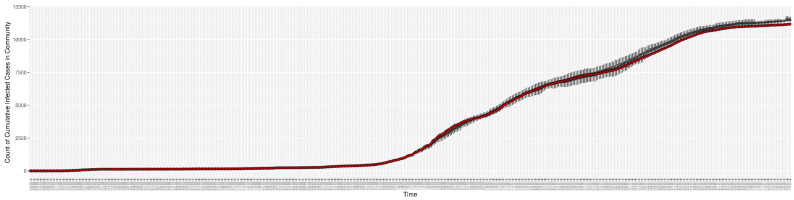
Cumulative reported infective cases in the community between particle filtering model results (boxplot) and empirical data (superimposed red scatterplot).

**Figure 5 ijerph-21-00193-f005:**
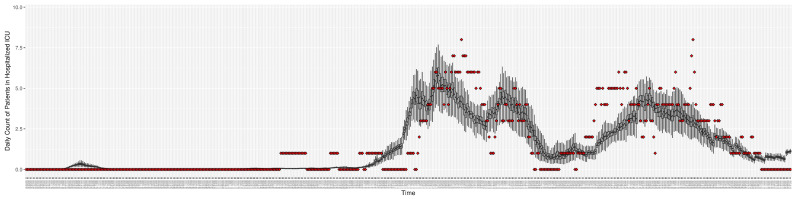
Daily count of COVID-19 patients hospitalized in the ICU between particle filtering model results (boxplot) and empirical data (superimposed red scatterplot).

**Figure 6 ijerph-21-00193-f006:**
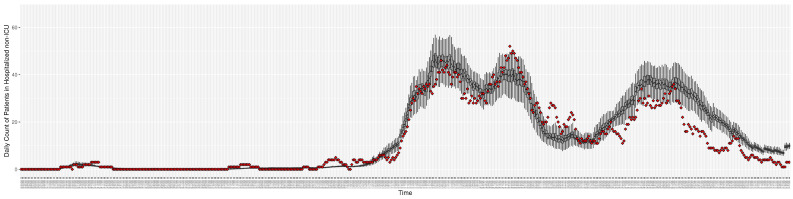
Daily count of COVID-19 patients hospitalized but not in the ICU between particle filtering model results (boxplot) and empirical data (superimposed red scatterplot).

**Figure 7 ijerph-21-00193-f007:**
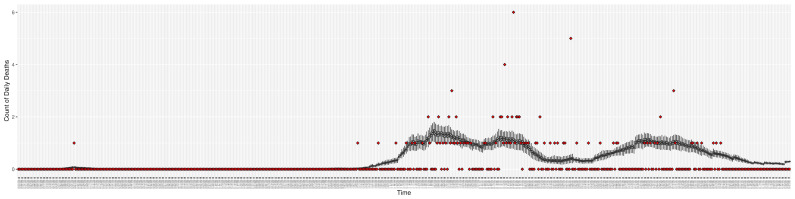
Daily count of COVID-19 hospitalized deaths between particle filtering model results (boxplot) and empirical data (red scatter plot).

**Figure 8 ijerph-21-00193-f008:**
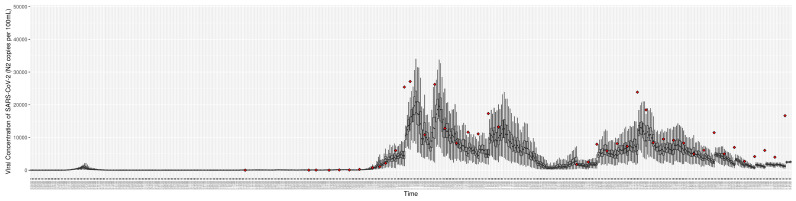
Daily wastewater viral concentration of SARS-CoV-2 (N2 copies per 100 mL) between particle filtering model results (boxplot) and empirical data with missing days (red scatter plot).

**Figure 9 ijerph-21-00193-f009:**
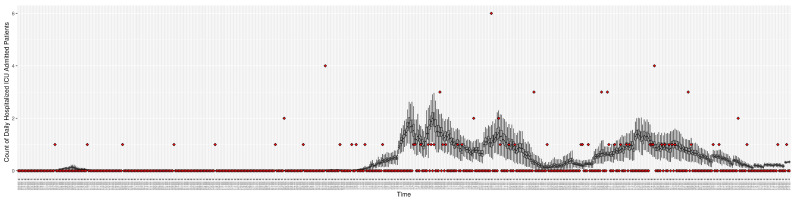
Daily count of hospitalized ICU-admitted patients between particle filtering model results (boxplot) and pseudo-empirical synthetic data (superimposed red scatterplot).

**Figure 10 ijerph-21-00193-f010:**
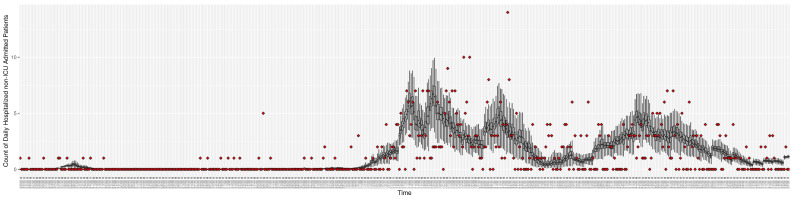
Daily count of hospitalized non-ICU-admitted patients between particle filtering model results (boxplot) and pseudo-empirical synthetic data (superimposed red scatterplot).

**Figure 11 ijerph-21-00193-f011:**
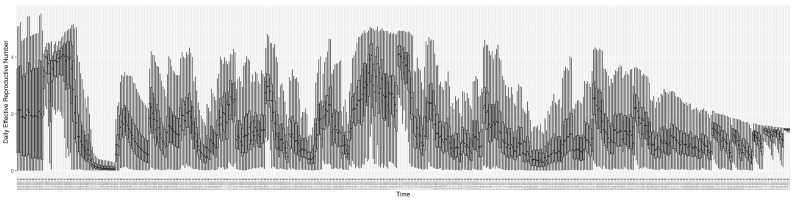
The daily estimated effective reproductive number.

**Figure 12 ijerph-21-00193-f012:**
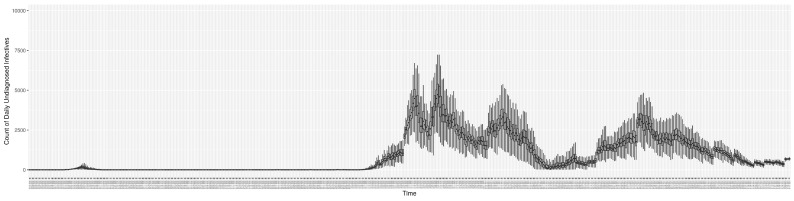
The daily estimated count of undiagnosed infectives.

**Figure 13 ijerph-21-00193-f013:**
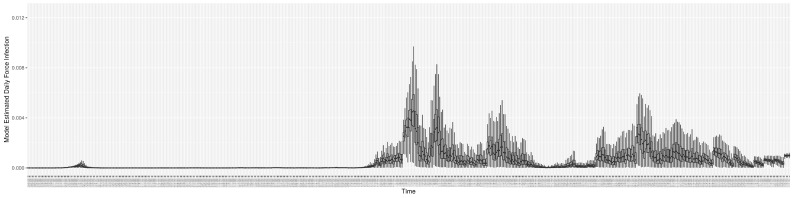
The daily estimated force of infection.

**Figure 14 ijerph-21-00193-f014:**
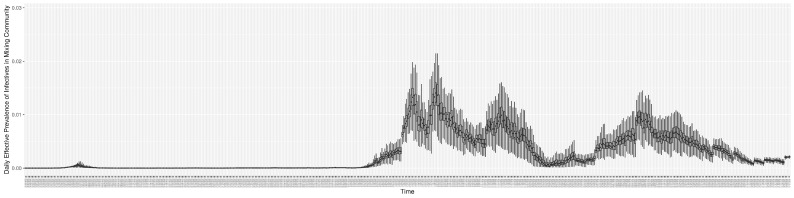
The daily estimated effective prevalence of infectives in the mixing community.

**Figure 15 ijerph-21-00193-f015:**
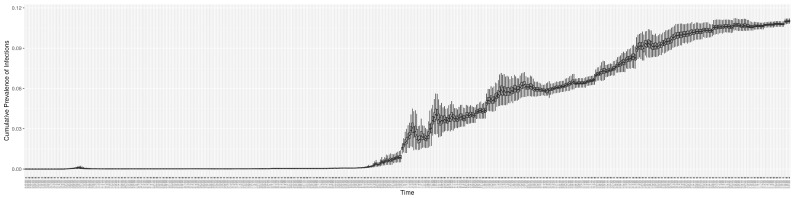
The daily estimated prevalence of cumulative infection.

**Figure 16 ijerph-21-00193-f016:**
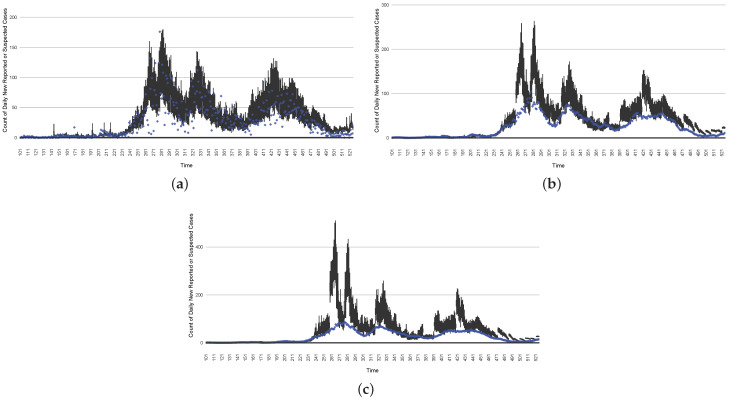
The projection results (boxplot) of the daily reported cases compared with empirical data (the blue points, not incorporated in the model). (**a**) The next-day projection results. (**b**) The 7-day time-window-averaged projection results versus corresponding time-window-averaged empirical data. (**c**) The 14-day time-window-averaged projection results versus corresponding time-window-averaged empirical data.

**Figure 17 ijerph-21-00193-f017:**
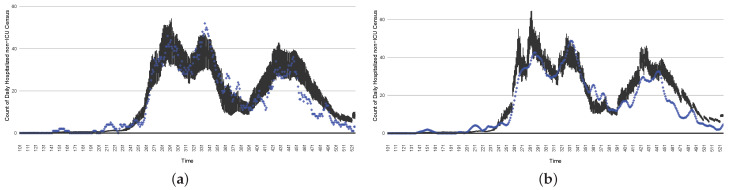
The projection results (boxplot) of the daily count of patients in the non-ICU compared with the empirical data (the blue points, not incorporated in the model). (**a**) The next-day projection results. (**b**) The 7-day time-window-averaged projection results versus corresponding time-window-averaged empirical data. (**c**) The 14-day time-window-averaged projection results versus corresponding time-window-averaged empirical data.

**Figure 18 ijerph-21-00193-f018:**
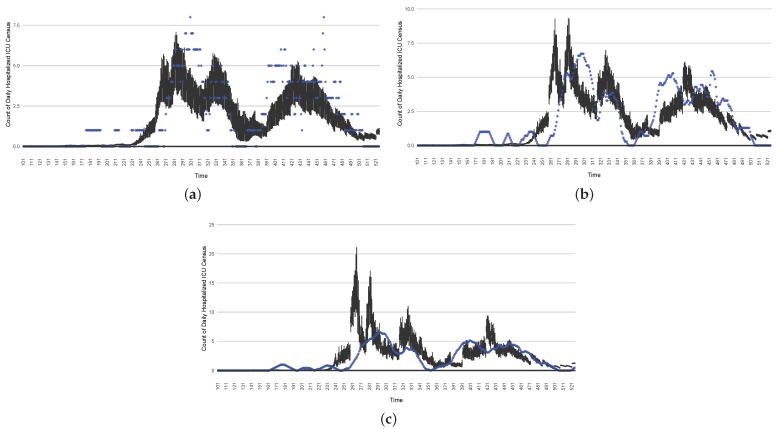
The projection results (boxplot) of the daily count of patients hospitalized in the ICU compared with the empirical data (the blue points, not incorporated in the model). (**a**) The next-day projection results. (**b**) The 7-day time-window-averaged projection results versus corresponding time-window-averaged empirical data. (**c**) The 14-day time-window-averaged projection results versus corresponding time-window-averaged empirical data.

**Figure 19 ijerph-21-00193-f019:**
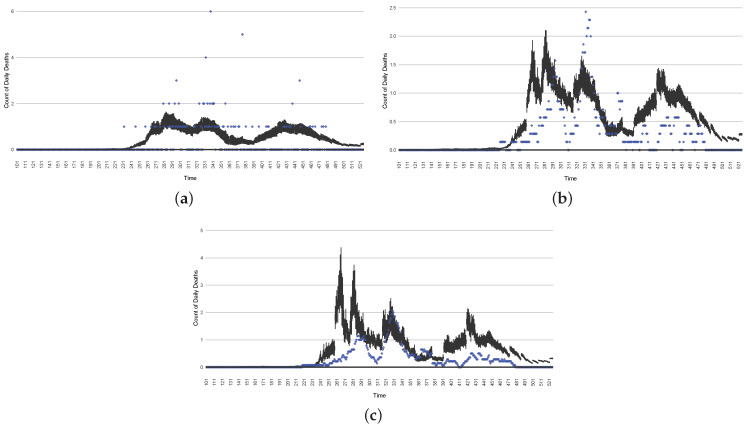
The projection results (boxplot) of the daily count of deaths compared with the empirical data (the blue points, not incorporated in the model). (**a**) The next-day projection results. (**b**) The 7-day time-window-averaged projection results versus corresponding time-window-averaged empirical data. (**c**) The 14-day time-window-averaged projection results versus corresponding time-window-averaged empirical data.

**Figure 20 ijerph-21-00193-f020:**
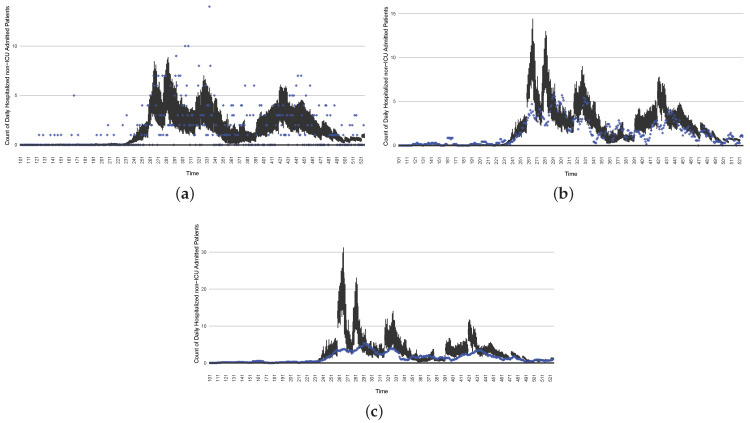
The projection results (boxplot) of the daily count of non-ICU-admitted patients with empirically mimicking synthetic data (the blue points, not incorporated in the model), with the latter being employed to preserve confidentiality. (**a**) The next-day projection results. (**b**) The 7-day time-window-averaged projection results versus corresponding time-window-averaged synthetic data. (**c**) The 14-day time-window-averaged projection results versus corresponding time-window-averaged synthetic data.

**Figure 21 ijerph-21-00193-f021:**
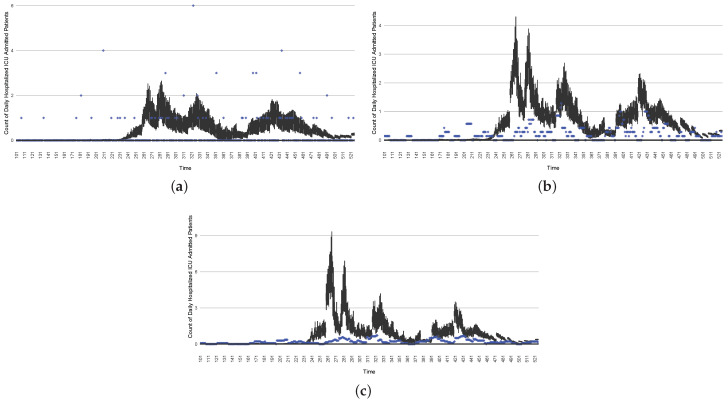
The projection results (boxplot) of the daily count of ICU-admitted patients with the empirically mimicking synthetic data (the blue points, not incorporated in the model), with the latter being employed to preserve confidentiality. (**a**) The next-day projection results. (**b**) The 7-day time-window-averaged projection results versus corresponding time-window-averaged synthetic data. (**c**) The 14-day time-window-averaged projection results versus corresponding time-window-averaged synthetic data.

**Figure 22 ijerph-21-00193-f022:**
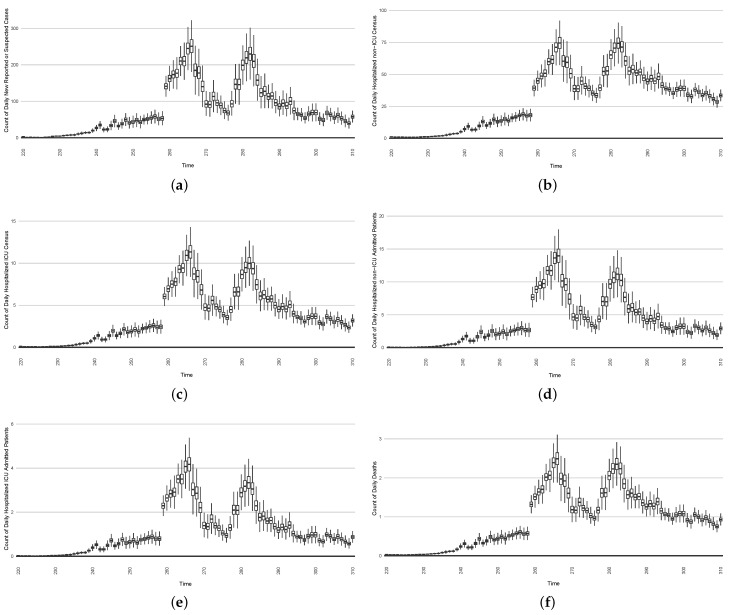
Model-based projections of the effects, for each successive day, of the average outcome of an intervention reducing the effective contact rate over the next 14 days, starting on that day.

**Figure 23 ijerph-21-00193-f023:**
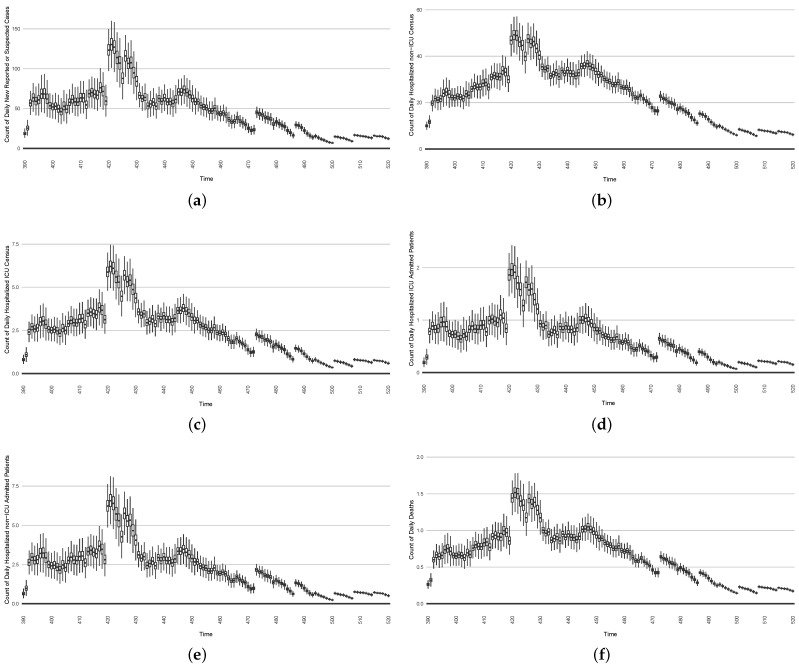
Model-based projections of COVID-19 for the next 14 days’ average results when simulating an outbreak-response immunization campaign. This is realized by characterizing a stylized elevated vaccine-induced protection level among 50% of the population.

**Table 1 ijerph-21-00193-t001:** Table of constant parameters.

Parameters	Description	Value	Source	Unit
$\rho_{U}$	Relative mixing rate amongst undiagnosed symptomatics	0.6	[39]${\;}^{1}$	1
$\rho_{D}$	Relative mixing rate amongst diagnosed in community	0.36	[39]${\;}^{2}$	1
$Ex_{D}$	Daily travel imported case count of diagnosed	Surveillance data	SHA primary data	Persons/Day
$E_{Vacc1}$	Daily count of persons administered the first-dose vaccination	Surveillance data	SHA primary data	Persons/Day
$E_{Vacc2}$	Daily count of persons administered the second-dose vaccination	Surveillance data	SHA primary data	Persons/Day
$V_{t}$	Daily count of persons undergoing PCR (nasopharyngeal swab)-based testing	Surveillance data	SHA primary data	Persons/Day
$V_{HICU}$	Daily count of COVID-19 patients admitted into the ICU	Surveillance data	SHA primary data	Persons/Day
$V_{HNICU}$	Daily count of COVID-19 patients admitted into the non-ICU	Surveillance data	SHA primary data	Persons/Day
$f_{S}$	Fraction of arriving symptomatics identified upon arrival	1/3	expert estimation	1
$f_{HICU}$	Fraction of admitting ICU among hospitalized patients	0.23	SHA primary data	1
$t_{E}$	Mean latent period	2.9	PHAC data	Day
$t_{I}$	Mean incubation period following infectivity	2.72	[40]	Day
$t_{I_{Y}}$	Mean time to develop or avoid complications	6.0	[41]	Day
$t_{R}$	Mean recovery time following symptoms	9.5	PHAC data	Day
$t_{H}$	Mean duration of hospital stay for non-ICU patients before recovery	12.0	SHA primary data	Day
$t_{ICU}$	Mean duration of ICU stay before move to hospital wards, discharge, or death	6.0	SHA primary data	Day
$t_{NICU}$	Mean duration of non-ICU stay before death	4.57	SHA primary data	Day
$f_{pA}$	Fraction of persistent asymptomatics	0.4	[42]	1
$\phi_{ICU}$	Case fatality rate amongst ICU patients	0.45	SHA primary data	1
$\phi_{NICU}$	Case fatality rate for cases not requiring ICU care	0.08	SHA primary data	1
$e_{1}$	Vaccine efficacy for dose 1	0.8	[31]	1
$e_{2}$	Vaccine efficacy for those completing two-doses primary series	0.95	[31]	1
γ	Ratio of model shedding measure to viral concentration in wastewater	10.374	PMCMC model [38]	copies/100 mL/Person
$w_{E}$	Viral shedding weight in exposed stage ($E_{U}$)	0.2	[37]	1
$w_{IA}$	Viral shedding weight in presymptomatic stage ($I_{AU}$, $I_{AD}$)	0.5	[37]	1
$w_{IY}$	Viral shedding weight in early symptomatic stage with complications and cotemporal stages of oligosymptomatic infection ($I_{A2U}$, $I_{A2D}$, $I_{YU}$, $I_{YUD}$, $H_{ICU}$, $H_{NICU}$)	0.2	[37]	1
$w_{IN}$	Viral shedding weight in symptomatic stage (absent complications) and cotemporal stage of oligosymptomatic infectives ($I_{A3U}$, $I_{A3D}$, $I_{YNU}$, $I_{YND}$)	0.1	[37]	1
$\beta_{T}$	Upper limit on fraction of infectives found by active testing	1.0	Reflective of full extent of unit range	1

^1^ We assume that 50% undiagnosed symptomatics reduce their contacts to 20% of normal, while the rest carry on as normal, and $\rho_{U}$ = 0.5 × 0.2 + (1 − 0.5) × 1 = 0.6. ^2^ We assume most diagnosed individuals in community totally isolated themselves but 20% not at all, and $\rho_{D}$ = 0.8 × 0.2 + (1 − 0.8) × 1 = 0.36.

**Table 2 ijerph-21-00193-t002:** Table of dynamic parameters.

Parameters	Meaning	Min (a)	Max (b)	STD	Unit
$C_{\beta }$	Transmission contact rate	0	0.4918 ${\;}^{1}$	10.0	Persons/Day
$f_{H}$	Fraction of symptomatic individuals who proceed on to require hospitalization	0.04	0.06	0.1	1
$f_{Y}$	Fraction of undiagnosed symptomatics who proceed on to seek care but who are not hospitalized	0.1	0.821	0.5	1
$\alpha_{t}$	A measure of test efficiency	0.01	0.25	5	1

${\;}^{1}$ Equivalent to a basic reproductive number $R_{0}=6$.

**Table 3 ijerph-21-00193-t003:** Table of sub-likelihood functions.

Likelihood Name	Empirical Dataset	Model Value	Mathematical Form
$L_{NewReportedEndogenousCases}$	New reported endogenous COVID-19 cases	$V_{P}+\frac{I_{YU}\left(f_{H}+f_{Y}\right)}{t_{I_{Y}}}$	Negative Binomial
$L_{CumulativeReportedEndogenousCases}$	Cumulative reported endogenous COVID-19 cases	$\int \left(V_{P}+\frac{I_{YU}\left(f_{H}+f_{Y}\right)}{t_{I_{Y}}}\right)$	Negative Binomial
$L_{CumulativeICUAdmission}$	Cumulative hospitalized ICU admission patients	$\int dH_{ICU}$	Negative Binomial
$L_{CumulativeNICUAdmission}$	Cumulative hospitalized non-ICU admission patients	$\int dH_{NICU}$	Negative Binomial
$L_{ICUCensus}$	Daily hospitalized ICU census patients	$H_{ICU}$	Negative Binomial
$L_{NICUCensus}$	Daily hospitalized non-ICU census patients	$H_{NICU}$	Negative Binomial
$L_{CumulativeDeaths}$	Cumulative COVID-19 deaths	*D*	Negative Binomial
$L_{ViralConcentration}$	Measured concentration of SARS-CoV-2 virus in wastewater	$\gamma [w_{E}E_{U}+w_{IA}\left(I_{AU}+I_{AD}\right)+\;w_{IY}\left(I_{A2U}+I_{A2D}+I_{YU}+I_{YUD}+\;H_{ICU}+H_{NICU}\right)+w_{IN}\left(I_{A3U}+\;I_{A3D}+I_{YNU}+I_{YND}\right)]$	Gamma Distribution

**Table 4 ijerph-21-00193-t004:** Table of discrepancies (normalized root mean square error (NRMSE)) of all empirical datasets compared with model estimated results (with 5 realizations).

Dataset	Mean	95% Confidence Interval
Count of daily reported cases	0.8429	(0.8343, 0.8515)
Cumulative reported cases	0.2750	(0.2558, 0.2943)
Cumulative death cases	0.4981	(0.4842, 0.5121)
Daily virus concentration in wastewater	0.5734	(0.5511, 0.5957)
Cumulative hospitalized non-ICU admissions	0.1306	(0.1223, 0.1389)
Cumulative hospitalized ICU admissions	0.5372	(0.5313, 0.5431)
Daily hospitalized non-ICU census	0.4181	(0.4117, 0.4245)
Daily hospitalized ICU census	0.6545	(0.6492, 0.6598)
Sum of total	3.9300	(3.8980, 3.9617)

**Table 5 ijerph-21-00193-t005:** Discrepancies for 1-day, 7-day, and 14-day projection run projected data against empirical datasets.

Dataset	Mean Projection Discrepancy		
	1-Day	7-Day	14-Day
Count of daily reported cases	0.7051	0.8301	0.9433
Cumulative reported cases	0.1591	0.1636	0.1769
Cumulative death cases	0.4098	0.4164	0.4293
Cumulative hospitalized non-ICU admissions	0.1617	0.1582	0.1734
Cumulative hospitalized ICU admissions	0.8705	0.8734	0.8838
Daily hospitalized non-ICU census	0.7131	0.7506	0.8308
Daily hospitalized ICU census	1.1541	1.1846	1.2364
Sum of total	4.1734	4.3767	4.6738

## Data Availability

The observed datasets were obtained under confidentiality agreement and within the authors’ secondment to public Saskatchewan Health Authority (SHA). Publication of this data would both violate the confidentiality agreement with the SHA and be contrary to the rules set down by the Saskatchewan Ministry of Health, to which the team also reported. However, noisified versions of all of the observed data are all plotted out (the red dots in the boxplots) in the paper, and the readers may estimate the values of the observed data based on the figures in the paper.

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
