# Peer review of "Real-Time Epidemiology and Acute Care Need Monitoring and Forecasting for COVID-19 via Bayesian Sequential Monte Carlo-Leveraged Transmission Models"

_ijerph, 2024, doi:10.3390/ijerph21020193_

Round 1

Reviewer 1 Report (New Reviewer)

Comments and Suggestions for Authors

The paper provides a comprehensive and innovative approach to COVID-19 epidemiology modeling. It addresses significant challenges in public health decision-making during the pandemic, particularly the need for models that can adapt to evolving data. The integration of various data sources, including municipal wastewater, adds a unique dimension to the study. However, the paper could benefit from more detailed discussions on model evolution, handling of emerging variants, and robustness against data variability.  I read this paper with great interests and have few comments hope can improve the paper. 

1.         The model’s ability to adapt to new data trends is emphasized. Could you discuss the specific mechanisms by which the model adapts to changing trends in COVID-19 data?

2.         The paper mentions the use of weighted shedding models based on the stage of infection. Could more details be provided on how these weights were determined and validated?

3.         How does the model handle variations in data quality and availability, especially in different geographic regions or over time?

4.         The paper does not extensively discuss the evolution of the model or its adaptations to accommodate new COVID-19 variants. Given the rapidly changing nature of the pandemic, how might the model's effectiveness be impacted by emerging variants?

5.         While the integration of wastewater surveillance data is innovative, it may be challenging to generalize this approach to regions without advanced wastewater monitoring infrastructure.

6.         The model’s reliance on accurate and timely data inputs could be a limiting factor, especially in regions where testing and reporting may be inconsistent.

7.         The literature review looks insufficient. I suggest to include these (not limited to):

a.         Koyama, S., Horie, T., & Shinomoto, S. (2021). Estimating the time-varying reproduction number of COVID-19 with a state-space method. PLoS Computational Biology, 17(5), e1008679. doi: 10.1371/journal.pcbi.1008679

b.          Wang, Q., Zhou, Y., & Chen, X. (2021). A vector autoregression prediction model for covid-19 outbreak. arXiv preprint arXiv:2102.04843.

Author Response

Reviewer 2 Report (New Reviewer)

Comments and Suggestions for Authors

The paper under review presents a comprehensive study on the design and multi-year deployment of a production-quality particle filter model used in public health decision-making during the COVID-19 pandemic. The model employs particle filtering dynamic (transmission) modeling and integrates health system and wastewater data to provide updated situational analyses and short-term forecasts. While the paper offers significant insights, there are several key areas where improvements and further research are recommended.

Drawbacks:

1. The most critical limitation is the model's inadequate characterization of the differential impact of vaccination on protection from infection versus protection from severe disease and death. The model only assesses vaccination's impact on transmission, neglecting the vaccinated individuals' reduced risk of hospitalization, ICU admission, and death post-infection.

2. The model assumes the presence of a single COVID-19 variant at a time, which limits its applicability in multi-variant scenarios, particularly in analyzing variant cross-reactivity and immunological protection.

3. The model applies to an aggregate population without considering heterogeneity. Stratification by age group, geographical location, and risk behavior would provide more nuanced insights, especially considering significant rural-urban disparities in vaccination and behavior.

4. The model does not account for the waning of natural and vaccine-induced immunity, a crucial aspect of the pandemic's dynamics.

5. While the paper discusses integrating wastewater data and health system data, it lacks a detailed explanation of the methodologies and challenges involved, especially at a larger scale.

Recommendations:

1. Modifying the model to differentiate between the effects of vaccination on transmission, infection, and severe outcomes is recommended. This includes incorporating mechanisms to retain vaccination status post-infection.

2. Enhance the model to simulate multiple variants simultaneously, allowing for studying competition and cross-reactivity among different strains.

3. Introduce stratification by age, geographic location, and other relevant factors to capture the diverse impact of COVID-19 across different population segments.

4. Update the model to include the dynamics of waning immunity over time for both natural and vaccine-induced immunity.

5. Provide a more detailed methodology for integrating and analyzing diverse data sources, such as health system and wastewater surveillance data, especially in different geographic and demographic settings.

6. It is recommended to extend the model to study the impact of successive booster vaccines, which is becoming increasingly relevant with the ongoing evolution of the virus.

7. Considering the model's complexity, exploring more sophisticated computational techniques and frameworks would be beneficial for handling the increased data volume and model complexity.

8. Conduct thorough validation exercises and sensitivity analyses to assess the robustness of the model under varying assumptions and scenarios.

In summary, while the paper provides valuable insights into particle filtering in pandemic modeling, addressing these drawbacks and incorporating the recommended enhancements will significantly improve its applicability and accuracy in public health decision-making.

Round 2

Reviewer 1 Report (New Reviewer)

Comments and Suggestions for Authors

No further comments from me.  Good luck!

Reviewer 2 Report (New Reviewer)

Comments and Suggestions for Authors

The authors have considered the reviewers' comments and recommendations.

This manuscript is a resubmission of an earlier submission. The following is a list of the peer review reports and author responses from that submission.

Round 1

Reviewer 1 Report

Comments and Suggestions for Authors

1. Abstract is very long. Reduce it to 1/3rd of the current text. Usually, no theory or motivation is included in the abstract. Similarly, the novelty of the proposed approach should be discussed.

2. Overall, the paper is too long and most of the things are too basic. With such a long paper, the readability is not assured. Also, the quality of the content is not guaranteed. Extensively reduce the length of the paper (it should be around 12-15 pages).

3. The introduction is sort of messey. Right after the 3rd paragraph, the authors start discussing the proposed approach. It is expected to cover the domain in depth and breadth in the introduction section before discussing the proposed approach.  Introduction should be reasonably extended.

4. The quality of writing requires serious considerations. The sentences are too long e.g., lines 52-55, 55-59, 60-66, 67-68, 72-77. Serious efforts should be made to improve the quality of writing. Take help from native English speakers in proofreading and reviewing the article.

5. I have serious concerns with respect to the choice of algorithm (Naive Bayes). The proposed transmission model is more or less a heuristics model (empirical data), in which takes different parameters as input such as natural histories  of infection, diagnosis status, infective and recovered (diagnosed, undiagnosed))population etc. Given such a rich set of parameters, it is unclear why the authors have chosen a heuristics based, empirical data dependent Bayesian machine learning algorithm of particle filtering when there are many better supervised and unsupervised learning algorithms available. It's a well known and well established fact that Naive Bayes heavily rely on a priori knowledge (which should be reasonably correct). They also face the so-called 'zero probability problem' and their performance is known to degrade with imbalanced data. Similarly, NBs assume all parameters as independent which rarely happens especially when considering the data used in the proposed work. Just because a model is predicting reasonably does not guarantee that it is the best possible model. I will suggest the authors to look into better ML methods (e.g. NN, SVM, RF) or DL models (LSTMs, RNNs or CNNs + RNNs) etc. Authors should present a comparison of NB, ML and/or DL algorithms along with a strong rationale in support of their choice. The work in the present form does not have any strong rationale on why NB is preferred.

6. All of the equations should be properly cited. If any modification is made in an existing equation, the modified part should be highlighted with a different color.

7. Sample data should be included for the readers so that they can get an idea of the type of data being recorded/used.

8. All of the graphs have readability issues (especially x-axis). Similarly the units should be indicated on the y-axis. Instead of plotting the whole graph, the author should only plot the interesting parts. Full scale graphs can be moved to the appendix section.

9. Figs 16,18, 19, 20, 21 do not show a good fit. Qualitative discussion should be included.

Reviewer 2 Report

Comments and Suggestions for Authors

1. Fig 3 to Fig 15, need X-axis details

2. Instead of providing multiple box plots, give some important typical  box  plots as a  result 

Reviewer 3 Report

Comments and Suggestions for Authors

The paper describes the design, implementation, and use of a particle filtered COVID-19 compartmental model that allows for continuous updating of state estimates in light of unfolding time series data. The authors argue that this approach provides a more reliable and accurate depiction of the evolving underlying epidemiology and acute care demand, which can support better decision-making around rolling back and re-instituting measures, initiating surge planning, and issuing public health advisories. The paper demonstrates the effectiveness of the model through its multi-year daily use for public health and clinical support decision-making in Canada.

Overall, the paper presents a compelling case for the use of machine learning algorithms, such as particle filtering, to continuously update and improve transmission models in response to changing epidemiological trends. The use of diverse data streams, including test volumes and positivity rates, hospital census and admissions flows, daily counts of vaccinations administered, and measured concentrations of SARS-CoV-2 in wastewater, enables the model to provide a more comprehensive and accurate depiction of the evolving situation on the ground. The probabilistic nature of the model also allows for the evaluation of trade-offs between potential intervention scenarios, which can further support evidence-based decision-making.

The paper is well-written and provides a detailed description of the model design and implementation, as well as its use in practice as an effective reporting tool. The use of a scripting pipeline that permits a fully automated reporting pipeline is particularly noteworthy and demonstrates the potential scalability of this approach for use in other jurisdictions. Overall, I would recommend this paper for publication, as it presents a valuable contribution to the field of COVID-19 transmission modeling and public health decision-making.

However, the manuscript could be improved by providing more information on the specific data sources used to inform the model and how the data is collected, processed, and validated. Additionally, it would be helpful to provide more detail on how the model's probabilistic state and parameter estimates are used to inform public health decision-making, including specific examples. The literature review is insufficent. some milestone works on the application of Bayesian machine learning should be reviewed, such as “Towards trustworthy machine fault diagnosis: A probabilistic Bayesian deep learning framework”, etc.